# Target protein identification in live cells and organisms with a non-diffusive proximity tagging system

Yingjie Sun[1†], Changheng Li[1†], Xiaofei Deng[2], Wenjie Li[1], Xiaoyi Deng[1], Weiqi Ge[1], Miaoyuan Shi[1], Ying Guo[1], Yanxun V Yu[1,3]*, Hai-bing Zhou[2,4]*, Youngnam N Jin[1,3]*

[1]Department of Neurology, Medical Research Institute, Zhongnan Hospital of Wuhan University, Wuhan University, Wuhan, China; [2]Department of Hematology, Zhongnan Hospital of Wuhan University, School of Pharmaceutical Sciences, Wuhan University, Wuhan, China; [3]Frontier Science Center for Immunology and Metabolism, Wuhan University, Wuhan, China; [4]State Key Laboratory of Virology, Hubei Province Engineering and Technology Research Center for Fluorinated Pharmaceuticals, Key Laboratory of Combinatorial Biosynthesis and Drug Discovery (Wuhan University), Ministry of Education, Frontier Science Center for Immunology and Metabolism, Wuhan University, Wuhan, China

*For correspondence:
yanxunyu@whu.edu.cn (YVY);
zhouhb@whu.edu.cn (H-bZ);
youngnam_jin@whu.edu.cn (YNJ)

†These authors contributed equally to this work

Competing interest: The authors declare that no competing interests exist.

## eLife Assessment

The study presents **important** findings that reveal SEPHS2 and VPS37C as new potential drug targets for dasatinib and hydroxychloroquine respectively in addition to confirming known targets of these drugs. The evidence provided is **compelling** as observed in the methods, data, and analyses. This article will be of great interest to chemical biologists, biochemists, and scientists in drug discovery and diagnostics.

**Abstract** Identifying target proteins for bioactive molecules is essential for understanding their mechanisms, developing improved derivatives, and minimizing off-target effects. Despite advances in target identification (target-ID) technologies, significant challenges remain, impeding drug development. Most target-ID methods use cell lysates, but maintaining an intact cellular context is vital for capturing specific drug–protein interactions, such as those with transient protein complexes and membrane-associated proteins. To address these limitations, we developed POST-IT (Pup-On-target for Small molecule Target Identification Technology), a non-diffusive proximity tagging system for live cells, orthogonal to the eukaryotic system. POST-IT utilizes an engineered fusion of proteasomal accessory factor A and HaloTag to transfer Pup to proximal proteins upon directly binding to the small molecule. After significant optimization to eliminate self-pupylation and polypupylation, minimize depupylation, and optimize chemical linkers, POST-IT successfully identified known targets and discovered a new binder, SEPHS2, for dasatinib, and VPS37C as a new target for hydroxychloroquine, enhancing our understanding these drugs' mechanisms of action. Furthermore, we demonstrated the application of POST-IT in live zebrafish embryos, highlighting its potential for broad biological research and drug development.

## Introduction

The landscape of modern pharmaceutical medicine has significantly evolved with the advent of enhanced drugs, targeted therapies, and combination treatments, yet variability in patient outcomes and toxic responses remain prevalent. This unpredictability stems from a complex interplay of factors, including incomplete target identification, unanticipated off-target effects, intricate polypharmacology, and the influence of individual genetics and environmental factors, all of which collectively contribute to the high failure rate of new drug candidates. This highlights the critical need for accurate identification of protein targets (target-ID). Phenotype-based drug screening offers a promising strategy for discovering novel or superior bioactive molecules, yet precise target-ID within this approach stands as a major barrier to advancing drug development further (*Hughes et al., 2021*; *Vincent et al., 2022*). Target-ID is fundamental in unraveling the mechanistic pathways through which drugs exert their therapeutic effects, serving as a critical step in the refinement and development of targeted therapies. By enabling the precise tailoring of drugs to specific biological targets, target-ID significantly enhances drug specificity, efficacy, and safety, while also facilitating the repurposing of existing drugs and advancing our understanding of polypharmacology. Despite its critical importance, accurately identifying biological targets within the complex cellular environment of proteins and other molecules remains a challenge, often hindering progress from drug discovery to therapeutic application.

Significant advancements have been made in target-ID technologies, with each approach offering its own advantages and limitations (*Ha et al., 2021*). Affinity-based pull-down methods, traditionally employed for target-ID, are hindered by their dependence on strong binding interactions, potentially overlooking proteins that interact transiently or are present at low levels but are crucial for the drug's action. Additionally, affinity-based methods do not adequately preserve the intricate intracellular environment where these interactions occur. As an alternative, photo-affinity labeling (PAL) has emerged, utilizing analogs of bioactive molecules with photo-reactive groups to detect interactions in the natural cellular state. Despite its advantages, PAL has limitations such as nonspecific labeling, low photoreaction yields, orientation dependency, and in vivo incompatibility. In response, newer, modification-free approaches like stability of proteins from rates of oxidation (*West et al., 2008*), drug affinity-responsive target stability (*Lomenick et al., 2009*), cellular thermal shift assay (CETSA) (*Molina et al., 2013*), and thermal proteome profiling *Savitski et al., 2014* have been developed. These techniques rely on protein stabilization upon binding, offering benefits like avoiding the need for cumbersome synthesis of derivatives and preserving the cellular context (*Molina et al., 2013*; *Savitski et al., 2014*). However, these methods may have difficulty detecting subtle or little changes in protein stability and have limited applicability to live cell conditions, underscoring the ongoing need for continued development in target identification methodologies.

Proximity labeling (PL) systems such as APEX (*Martell et al., 2012*)/APEX2 (*Lam et al., 2015*), BioID (*Roux et al., 2012*), and TurboID (*Branon et al., 2018*) are rapidly emerging as effective tools for dissecting protein–protein or RNA–protein molecular interactions. While these methods offer significant advantages in mapping interactomes and networks, their diffusive labeling presents a limitation for accurate target-ID. Although certain developments such as the proximity-dependent target-ID system (PROCID) (*Kwak et al., 2022*) and biotin targeting chimera (BioTAC) (*Tao et al., 2023*) using TurboID and miniTurbo, respectively, have shown promise, the inherent diffusive nature still remains an obstacle to directly pinpointing target proteins, leading to increased potential for confusing results or necessitating extensive validation steps. Wells and colleagues developed a non-diffusive PL system for protein–protein interactions by fusing the E2-conjugating enzyme Ubc12 for NEDD8, a ubiquitin homolog, with a substrate binding domain/protein (*Zhuang et al., 2013*). This system was further adapted for target-ID by combining Ubc12 with a SNAP-tag for linkage to a specific small molecule, allowing Ubc12 to covalently attach a biotinylated NEDD8 derivative to adjacent target proteins (*Hill et al., 2016*). While effective for target-ID, it has limitations. Exogenous expression of NEDD8 can lead to unintended labeling by the ubiquitination machinery (*Enchev et al., 2015*), thus restricting its application in live cells. Consequently, this system is generally utilized in vitro with cell lysates. While capable of tagging weakly interacting proteins, the system does not utilize the intact intracellular context, which is crucial for certain interactions. Additionally, the possibility of NEDD8 polyneddylation may introduce bias in labeling efficiency and affinity pulldown. Furthermore, the typical neddylation conditions, occurring at 37°C for 18 hr, may cause protein instability and precipitation, although adjustments in incubation time or the addition of mild

detergents may alleviate these concerns. Hence, there is a pressing need to develop innovative target-ID methods that enhance existing techniques.

To overcome existing limitations and establish a streamlined approach for target-ID in live cells and animal models such as zebrafish, we envisioned a non-diffusive PL system that leverages a prokaryotic ubiquitin-like protein, orthogonal to the eukaryotic ubiquitin proteasome system, thereby ensuring specificity and minimizing cross-reactivity. We chose the proteasomal accessory factor A (PafA) as a ligase and the prokaryotic ubiquitin-like protein (Pup) as its cognate substrate. *Liu et al., 2018* had previously demonstrated its effectiveness as a PL system for studying protein–protein interactions in live cells. Here, we introduce a target-ID system, named Pup-On-target for Small molecule Target Identification Technology (POST-IT). This system integrates PafA with a HaloTag to attach a specific small molecule. After extensive optimization, we showed that POST-IT successfully identified many known targets and a novel binder, SEPHS2, for dasatinib in live cells. We then applied POST-IT for hydroxychloroquine (HCQ) and discovered VPS37C as a novel target, providing mechanistic insight into HCQ's action. Moreover, we have successfully demonstrated the application of POST-IT in live zebrafish embryos, highlighting its potential for broad usage in biological research.

## Results
### Design and optimization of POST-IT
To create a non-diffusive PL system suitable for universal and robust target-ID in live cells and animal models, we integrated PafA from *Corynebacterium glutamicum* (*Cglu*) as a non-diffusive PL effector with HaloTag as an anchor for small molecules of interest (*Figure 1*). PafA covalently transfers Pup to proximal prey proteins via direct contact on the ε-amino group of lysine residues (*Özcelik et al., 2012*; *Watrous et al., 2010*), a process known as pupylation. This approach offers an advantage due to PafA's inherent characteristic of promiscuous, low selectivity for the amino acid sequence (*Liu et al., 2018*; *Watrous et al., 2010*). We selected HaloTag over alternatives like SNAP-tag due to its superior live-cell labeling efficiency (*Erdmann et al., 2019*), readily accessible ligand surface (*Kang et al., 2017*; *Los et al., 2008*), and relatively simple chemical synthesis with a chloroalkane moiety (*Los et al., 2008*). To simplify our system for in vivo applications without requiring exogenous biotin, we fused Pup with either a streptavidin binding peptide (SBP) or a Strep-tag II (STII) as a handle to capture the targets, rather than using the commonly used biotin carboxyl carrier protein (BCCP). When a target protein binds to the compound linked to HaloTag, PafA is positioned in close proximity, allowing for the covalent attachment of Pup to the target's lysine residue. Pupylated targets can then be captured and identified by mass spectrometry.

To initiate the optimization of POST-IT, we reasoned that using smaller Pup substrates could minimize interference with the cellular functions of pupylated proteins. It has been shown that the N terminus of Pup can be removed without compromising its function, generating a short truncated protein with 28 amino acids from the C terminus (*Liu et al., 2018*). This truncated Pup protein will be referred to hereafter as short Pup (sPup). We fused sPup with either an SBP or a twin-STII (TS) tag at its N terminus and used HB-Pup as a control, which contains 6×His and BCCP at the N terminus of Pup (*Figure 1—figure supplement 1A*). HaloTag was introduced at the N terminus of PafA. In vitro pupylation tests with recombinant proteins showed that the fusion of HaloTag did not impair PafA activity based on self-pupylation (*Figure 1—figure supplement 1B*). Notably, both types of sPup substrates exhibited similar pupylation levels to the full-length Pup. Additionally, PafA retained its robust activity even at 20°C, a valuable feature for using this system in other animal models such as zebrafish and *Caenorhabditis elegans*, which naturally inhabit environments around this temperature. Moreover, we observed robust self-pupylation within 10 min at 37°C. However, extensive incubation resulted in the formation of high molecular weight Halo-PafA, indicating polypupylation of Pup substrates (*Figure 1—figure supplement 1C*).

Polypupylation refers to the addition of a new Pup onto a previously linked Pup on the target protein, akin to polyubiquitination observed with ubiquitin. In contrast, multipupylation involves multiple single pupylations at different positions on the target proteins. Polypupylation can lead to unwanted biases in pupylation or pull-down efficiency, whereas multipupylation is more likely to correlate with binding affinity or interaction frequency. To counter polypupylation, we mutated lysine residues to arginine in TS-sPup and SBP-sPup (*Figure 1—figure supplement 2A*). Since the

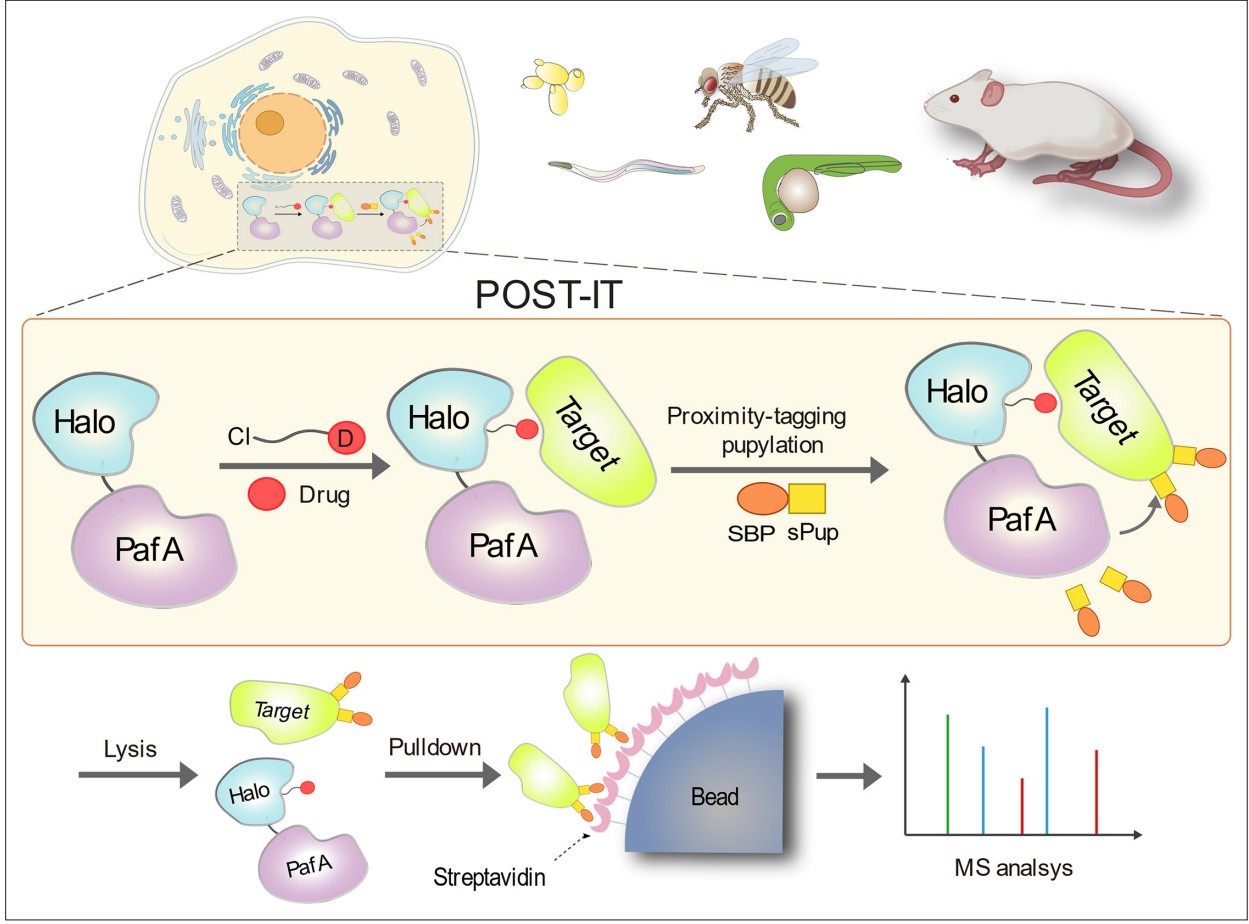

**Figure 1.** Schematic illustration of the POST-IT system for target identification. Upon the introduction of a drug-specific HaloTag ligand derivative, POST-IT tags target proteins with sPup, a tagging process known as pupylation, in live cells or organisms. Subsequently, these proteins are enriched and identified via mass spectrometry (MS) analysis.

The online version of this article includes the following source data and figure supplement(s) for figure 1:

**Figure supplement 1.** PafA fused with HaloTag exhibits high activity across a wide temperature range.

**Figure supplement 1—source data 1.** Uncropped and labeled gels of Coomassie blue staining for *Figure 1—figure supplement 1*.

**Figure supplement 1—source data 2.** Raw unedited gels of Coomassie blue staining for *Figure 1—figure supplement 1*.

**Figure supplement 2.** Lysine-mutated Pup substrates bypass polypupylation while retaining activities comparable to wild-type Pup.

**Figure supplement 2—source data 1.** Uncropped and labeled gels of Coomassie blue staining for *Figure 1—figure supplement 2*.

**Figure supplement 2—source data 2.** Raw unedited gels of Coomassie blue staining for *Figure 1—figure supplement 2*.

**Figure supplement 3.** Optimization of Halo-PafA.

**Figure supplement 3—source data 1.** Uncropped and labeled gel of Coomassie blue staining for *Figure 1—figure supplement 3*.

**Figure supplement 3—source data 2.** Raw unedited gel of Coomassie blue staining for *Figure 1—figure supplement 3*.

TS-tag contains two lysines, mutating sPup alone did not affect polypupylation (*Figure 1—figure supplement 2B and C*). However, further mutations of two lysine residues within the TS-tag, creating TS$^{K8R}$-sPup$^{K61R}$, completely abolished polypupylation in TS-sPup (*Figure 1—figure supplement 2D*). Interestingly, Halo-PafA with polypupylation branches, marked by yellow arrows, showed a distinct and slower migration pattern compared to multipupylation by TS$^{K8R}$-sPup$^{K61R}$ at 0.5 hr incubation. Similarly, modifying a lysine residue in the SBP-tag, either through mutation to SBP$^{K4R}$ or by deleting the N-terminal four amino acids to create sSBP, also completely prevented polypupylation, as shown in sSBP-sPup$^{K61R}$ and SBP$^{4KR}$-sPup$^{K61R}$ (*Figure 1—figure supplement 2E and F*). Notably, we observed enhanced levels of multipupylation with lysine-removed sPup substrates over time (*Figure 1—figure supplement 2D and F*). Together, these results demonstrate that lysine to arginine (KR) mutation

in sPup and modifications to the tags effectively eliminated polypupylation, while maintaining or enhancing the pupylation efficiency in a dose- and time-dependent manner.

In addition to polypupylation, we reasoned that the robust self-pupylation activity of Halo-PafA could diminish the pupylation of target proteins. To address this, we replaced all eight lysines with arginines in HaloTag, creating Halo$^{8KR}$-PafA (*Figure 1—figure supplement 3A*). Ts-sPup and TS-sPup$^{K61R}$ were chosen as sPup substrates for this experiment, although any Pup substrates could have been used. The levels of self-pupylation were assessed. The result showed that Halo$^{8KR}$-PafA mutant exhibited a marked decrease in self-pupylation, similar to PafA alone, indicating inherent self-pupylation in PafA (*Figure 1—figure supplement 3B*). Previous research reported that PafA from *Mycobacterium tuberculosis* (*Mtb*) possesses depupylase activity (*Jiang et al., 2018*), which may negatively impact pupylation levels of target proteins. Notably, an *Mtb* PafA mutant with a S119A mutation

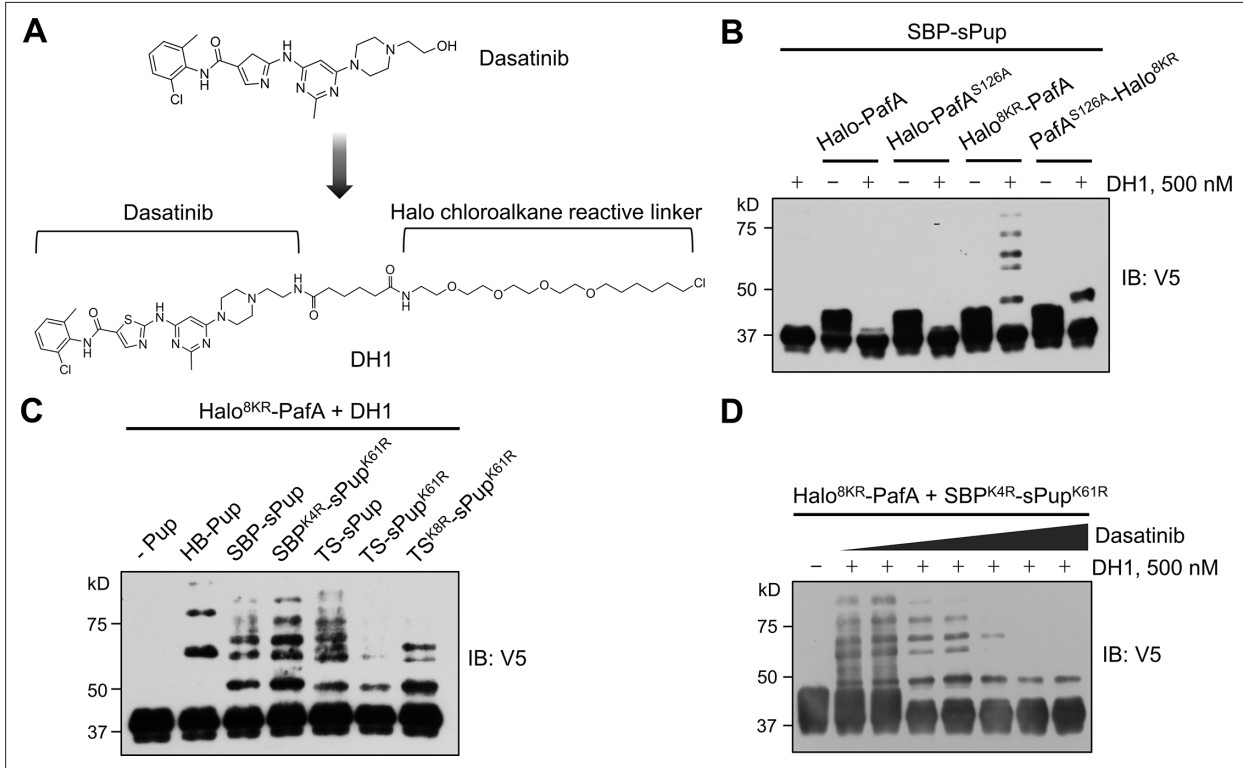

**Figure 2.** Optimized Halo-PafA and Pup substrates efficiently promote pupylation in vitro in a proximity-dependent manner. (**A**) The chemical structure of DH1, an HTL derivative of dasatinib. (**B**) Comparison of proximity-tagging by different Halo-PafA derivatives on in vitro pupylation. (**C**) Comparison of different Pup substrates on in vitro pupylation with 500 nM DH1. (**D**) Pupylation levels by DH1 decrease with an increasing amount of dasatinib as a competitor, ranging from 0.2 μM to 10 μM. (**B–D**) All reactions were conducted with 1 μM of a Halo-PafA derivative, 10 μM of one of the different Pup substrates, and 0.5 μM of a purified short SRC with a 2xV5 tag, SRC(247-536)-2xV5, at 37°C for 30 min. Pupylation levels of SRC were assessed by immunoblot (IB) using an anti-V5 antibody.

The online version of this article includes the following source data and figure supplement(s) for figure 2:

**Source data 1.** File containing labeled original western blots for *Figure 2*.

**Source data 2.** Original files for western blots displayed in *Figure 2*.

**Figure supplement 1.** Synthesis of dasatinib-Halo derivative 1 (DH1).

**Figure supplement 2.** Representative spectra of DH1.

**Figure supplement 3.** Halo$^{8KR}$-PafA demonstrates efficient labeling with HaloTag ligands.

**Figure supplement 3—source data 1.** File containing labeled original western blots for *Figure 2—figure supplement 3*.

**Figure supplement 3—source data 2.** Original files for western blots displayed in *Figure 2—figure supplement 3*.

**Figure supplement 4.** Optimized Halo-PafA and Pup substrates efficiently promote pupylation in vitro in a proximity-dependent manner.

**Figure supplement 4—source data 1.** File containing labeled original western blots for *Figure 2—figure supplement 4*.

**Figure supplement 4—source data 2.** Original files for western blots displayed in *Figure 2—figure supplement 4*.

showed significantly reduced depupylase activity with minimal changes in pupylase activity (*Jiang et al., 2018*). Based on this, we hypothesized that inhibiting depupylation activity might enhance pupylation. Surprisingly, a PafA mutant with an S126A substitution (analogous to S119A in *Mtb* PafA) exhibited substantially increased pupylation activity with reduced polypupylation, as evidenced by the decreased levels of high molecular weight bands and an increase in low molecular weight bands (*Figure 1—figure supplement 3A and B*).

## Validation of POST-IT with dasatinib in vitro

To assess the efficiency of our optimized Halo-PafA and Pup in pupylating target proteins for a small molecule, we synthesized DH1, a HaloTag ligand (HTL) derivative of dasatinib (*Figure 2A*; *Figure 2— figure supplement 1*; *Figure 2—figure supplement 2*), a well-characterized tyrosine kinase inhibitor used for leukemia treatment (*Rix et al., 2007*; *Shi et al., 2012*). Initially, we evaluated the functionality of Halo$^{8KR}$ in Halo$^{8KR}$-PafA. Fluorescence polarization (FP) assay demonstrates the rapid binding of the Halo-AF488 ligand to Halo$^{8KR}$-PafA, reaching a plateau in 15–20 min, consistent with a previous report (*Los et al., 2008*; *Figure 2—figure supplement 3A*). A Halo-biotin competition assay confirmed further that DH1 could be efficiently linked to Halo$^{8KR}$-PafA within 15 min (*Figure 2—figure supplement 3B and C*).

After demonstrating the functionalities of Halo$^{8KR}$ and DH1, we tested whether SRC, a known target of dasatinib, could be labeled by Halo-PafA in the presence of DH1. Surprisingly, robust pupulation of purified recombinant short SRC occurred exclusively with PafA harboring Halo$^{8KR}$ at its N terminus in the presence of DH1, whereas PafA with Halo$^{8KR}$ at the C terminus showed only modest labeling. This result highlights the importance of eliminating self-pupylation and the N-terminal positioning of HaloTag for effectively labeling the target protein (*Figure 2B*). We next examined the influence of KR mutations in sPup substrates. Both SBP$^{K4R}$-sPup$^{K61R}$ and TS$^{K8R}$-sPup$^{K61R}$ showed substantial levels of pupulation, especially at lower pupulation bands, suggesting increased multipupylation (*Figure 2C*). The pupulation of SRC by Halo$^{8KR}$-PafA with DH1 diminished with increasing concentrations of dasatinib, affirming the specificity of this system (*Figure 2D*). Notably, even with high concentration of SRC and Halo$^{8KR}$-PafA in the in vitro reaction, no pupulation was detected in the absence of DH1, indicating that the labeling by our PL system is non-diffusive, specific, and proximity-dependent.

Like SBP$^{K4R}$-sPup$^{K61R}$, we observed a significant increase in pupulation by PafA with the N-terminal, but not the C-terminal, Halo$^{8KR}$ when using TS$^{K8R}$-sPup$^{K61R}$ (*Figure 2—figure supplement 4A*) to a degree similar to that of SBP$^{K4R}$-sPup$^{K61R}$ (*Figure 2—figure supplement 4B*). Although TS$^{K8R}$-sPup$^{K61R}$ serves as an excellent substrate for PafA without causing polypupylation, the lysine mutations in the TS-tag led to a loss of binding to Strep-Tactin beads. Conversely, SBP$^{K4R}$ and sSBP retained their binding to streptavidin, as the core binding sequence of SBP does not require the 1–10 amino acids at the N-terminus (*Barrette-Ng et al., 2013*). Therefore, we decided to use the SBP-tag as a Pup substrate in the POST-IT system for further studies.

## Validation of POST-IT with dasatinib in cellulo

Having successfully verified target protein pupulation by Halo-PafA in vitro, we evaluated our PL system under cellular conditions to minimize artifacts from non-physiological environments such as cell lysates. To mimic drug–target interaction-induced pupulation in live cells and assess the potential of PafA as a proximity-tagging system for target-ID, we incorporated the rapamycin-induced interaction between FRB and FKBP into our PL system as this interaction between a small molecule and a protein is known to be highly specific and robust (*Figure 3—figure supplement 1A*). Rapamycin treatment led to robust pupulation of FKBP-EGFP by PafA fused with FRB in a Pup-dose dependent manner (*Figure 3—figure supplement 1B and C*), along with significant self-pupylation independent of rapamycin. Next, we investigated how KR mutations in SBP-sPup substrates affect pupulation efficiency in live cells. Both SBP-sPup and SBP$^{K4R}$-sPup$^{K61R}$ exhibited efficient pupulation (*Figure 3—figure supplement 1D*), while sSBP-sPup$^{K61R}$ failed to induce self-pupylation as well as pupulation of FKBP-EGFP, presumably due to its low expression or instability. Consequently, we chose SBP$^{K4R}$-sPup$^{K61R}$ for subsequent studies to avoid polypupylation, although SBP-sPup did not cause any detectable polypupylation in this case. Interestingly, the presence of rapamycin greatly reduced self-pupylation when using either SBP$^{K4R}$-sPup$^{K61R}$ or SBP-sPup, as shown in the lower panel of *Figure 3—figure supplement 1D*, suggesting that PafA constantly pupulates either itself or proximal target proteins.

**Table 1.** Identification of lysine residues for pupylation in PafA by mass spectrometry.

| Protein | Score | Coverage (%) | # PSMs* | Detected pupylated sequences | MH+ (Da) | Xcorr | Lysine | Rank of pupylation |
|---------|-------|--------------|---------|------------------------------|----------|-------|--------|--------------------|
| PafA | 607.055 | 89 | 1197 | | | | | |
| | | | 1 | RIMGIETEYGLTFVDGDSKK | 2502.21808 | 5.40 | | |
| | | | 2 | RIMGIETEYGLTFVDGDSKK | 2518.21299 | 5.35 | | |
| | | | 2 | IMGIETEYGLTFVDGDSKK | 2362.11188 | 3.71 | K29 | 3 |
| | | | 1 | IMGIETEYGLTFVDGDSKK | 2346.11697 | 3.46 | K30 | 5 |
| | | | 1 | MFRPIVEKYSSSNIFIPNGSR | 2702.32428 | 2.93 | K47 | 5 |
| | | | 1 | MAVDAEESLAK | 1406.64679 | 3.39 | | |
| | | | 2 | MAVDAEESLAKEDIAGQVYLFK | 2670.29672 | 3.86 | | |
| | | | 4 | MAVDAEESLAKEDIAGQVYLFK | 2686.29164 | 3.86 | K106 | 2 |
| | | | 5 | IHHPNPLDKGESFPLGYCISQR | 2808.35222 | 5.10 | | |
| | | | 3 | IHHPNPLDKGESFPLGYCISQR | 2809.33624 | 2.88 | K172 | 1 |
| | | | 1 | ILKAADTLGVPVTVDWMR | 2228.17436 | 3.61 | | |
| | | | 1 | ILKAADTLGVPVTVDWMR | 2244.16928 | 3.49 | K427 | 4 |

*PSM: peptide-spectrum match.

This underscores the importance of minimizing self-pupylation as a critical factor in developing for a better POST-IT system.

Given these findings, we sought to identify additional self-pupylation sites in PafA through mass analysis, uncovering six lysine residues susceptible to self-pupylation (*Table 1*). Focusing on the most frequently pupylated residue, we introduced an additional KR mutation at K172 (*Figure 3—figure supplement 2*), generating Halo[8KR]-PafA[S126A,K172R].

To further refine our PL system, we tested the effect of linker length between Halo[8KR] and PafA on SRC labeling in the presence of DH1. We observed enhanced labeling with a linker longer than 10 amino acids (*Figure 3A*), prompting us to select an 18-amino acid linker for further studies. We chose the 18-amino acid linker instead of the 40-amino acid linker for easier cloning and to lower the risk of DNA recombination from longer repeats. Additionally, a longer, flexible linker may behave like an intrinsically disordered protein (*Harmon et al., 2017*), which is an unwanted feature for target-ID. We then confirmed that the S126A and K172R mutations additively enhanced pupylation activity under cellular conditions as the higher molecular weight multipupylation bands were slightly but noticeably increased with these mutations compared to Halo[8KR]-PafA (*Figure 3B*). This optimized version of Halo-PafA, Halo[8KR]-PafA[S126A,K172R], in combination with a polypulation-free version of sPup, will be referred to as POST-IT. Unlike wild-type Halo-PafA, which displayed negligible labeling under cellular conditions, POST-IT exhibited significant labeling activity on both exogenous and endogenous SRC in the presence of DH1, generating POST-IT[DH1] (*Figure 3C–E*). Labeling intensity of short SRC increased with higher DH1 concentrations and was competitively inhibited by increasing concentrations of dasatinib (*Figure 3F and G*), demonstrating the specificity of POST-IT[DH1] in target labeling.

In our quest for a superior PafA variant, we investigated the potential of *Mtb* PafA as a PL system. However, we detected no labeling of short SRC by *Mtb* PafA, irrespective of the positioning of Halo[8KR] at the N- or C-terminus (*Figure 3—figure supplement 3*). In contrast, Halo[8KR]-PafA (*Cglu*) exhibited significant pupylation with sPup[K61R] substrates from both *Cglu* and *Mtb*, suggesting a lack of orthogonality among different prokaryotic PafA systems.

## Optimization of the linker of dasatinib-HTL (DH) derivatives

Encouraged by the successful labeling of the target protein by POST-IT[DH1] in live cells, we next explored its ability to identify known targets and potentially reveal novel binding proteins of dasatinib. To this end, we investigated whether the linker length or structure between dasatinib and the Halo chloroalkane moiety could influence labeling efficiency, as the linker's characteristics in biomolecules,

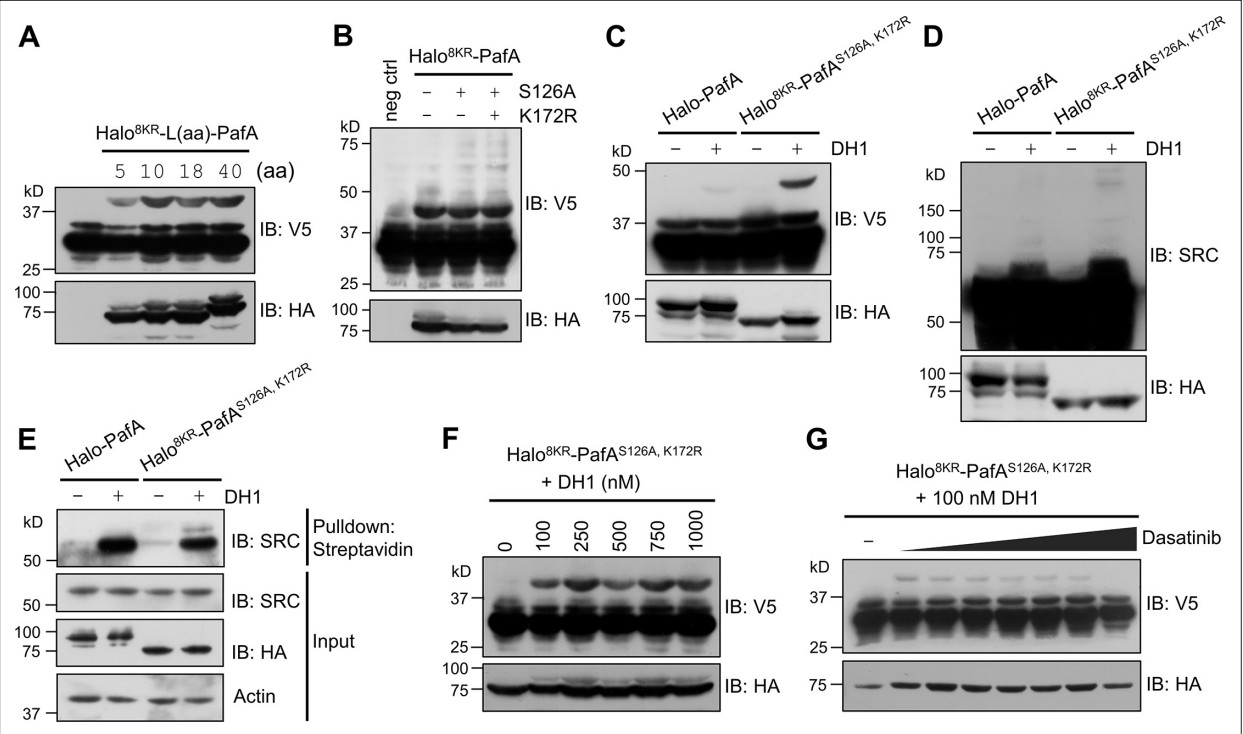

**Figure 3.** POST-IT can label target proteins in live cells. (**A**) Effect of linker length between Halo[8KR] and PafA on target-labeling. HEK293T cells were co-transfected with HA-Halo[8KR]-PafA containing the specified length of linker, SBP[K4R]-sPup[K61R], and SRC(247-536)-2xV5. (**B**) Introducing mutations S126A and K172R into PafA significantly enhances its proximity-tagging efficiency on target. (**C–E**) Comparison of labeling activity of Halo-PafA and Halo[8KR]-PafA[S126A,K172R], a form used for POST-IT. (**F, G**) POST-IT[DH1] mediates proximity-tagging in a DH1-dose dependent manner (**F**), an effect completely inhibited by competitive dasatinib, ranging from 0.025 to 25.6 μM (**G**). Pupylation levels of exogenously expressed short SRC (**C, F, G**), endogenous SRC (**D**), or pull-downed endogenous SRC (**E**) were assessed by immunoblot (IB) analysis using antibodies against V5-tag or SRC for exogenous or endogenous SRC, respectively. In all experiments, SBP[K4R]-sPup[K61R] was used for co-transfection, and cells were treated with 500 nM DH1, except in (**F, G**).

The online version of this article includes the following source data and figure supplement(s) for figure 3:

**Source data 1.** File containing labeled original western blots for *Figure 3*.

**Source data 2.** Original files for western blots displayed in *Figure 3*.

**Figure supplement 1.** Test of small molecule-induced proximity-tagging by PafA in HEK293T cells.

**Figure supplement 1—source data 1.** File containing labeled original western blots for *Figure 3—figure supplement 1*.

**Figure supplement 1—source data 2.** Original files for western blots displayed in *Figure 3—figure supplement 1*.

**Figure supplement 2.** Identification of self-pupylated lysine residues in PafA.

**Figure supplement 3.** PafA from *Cglu* outperforms PafA from *Mtb* in the POST-IT system in live cells.

**Figure supplement 3—source data 1.** File containing labeled original western blots for *Figure 3—figure supplement 3*.

**Figure supplement 3—source data 2.** Original files for western blots displayed in *Figure 3—figure supplement 3*.

including PROTAC (*Békés et al., 2022*), significantly impact their functionalities. Based on a previous report highlighting the enhanced performance of HTLs containing carbamate linkers (*Friedman Ohana et al., 2015*), we synthesized four additional DH derivatives (DH2-DH5) with varying linker lengths longer than that in DH1 (*Figure 4—figure supplements 1–6*). These new DH derivatives with carbamate linkers displayed significantly enhanced covalent binding to recombinant Halo[8KR]-PafA in vitro and to transiently expressed Halo[8KR]-PafA in intact cells, as measured by FP assays, compared to DH1 (*Figure 4A and B*). Importantly, the new DH derivatives also showed more robust pupylation of short SRC in vitro and endogenous SRC under cellular conditions (*Figure 4C and D*). DH5, containing the longest linker, was selected for further experiments due to its excellent efficiency. We then verified that POST-IT could efficiently pull down endogenous SRC in the presence of DH5, an effect attenuated by an excess of dasatinib (*Figure 4E*). Furthermore, we measured the inhibitory effect of DH5 on

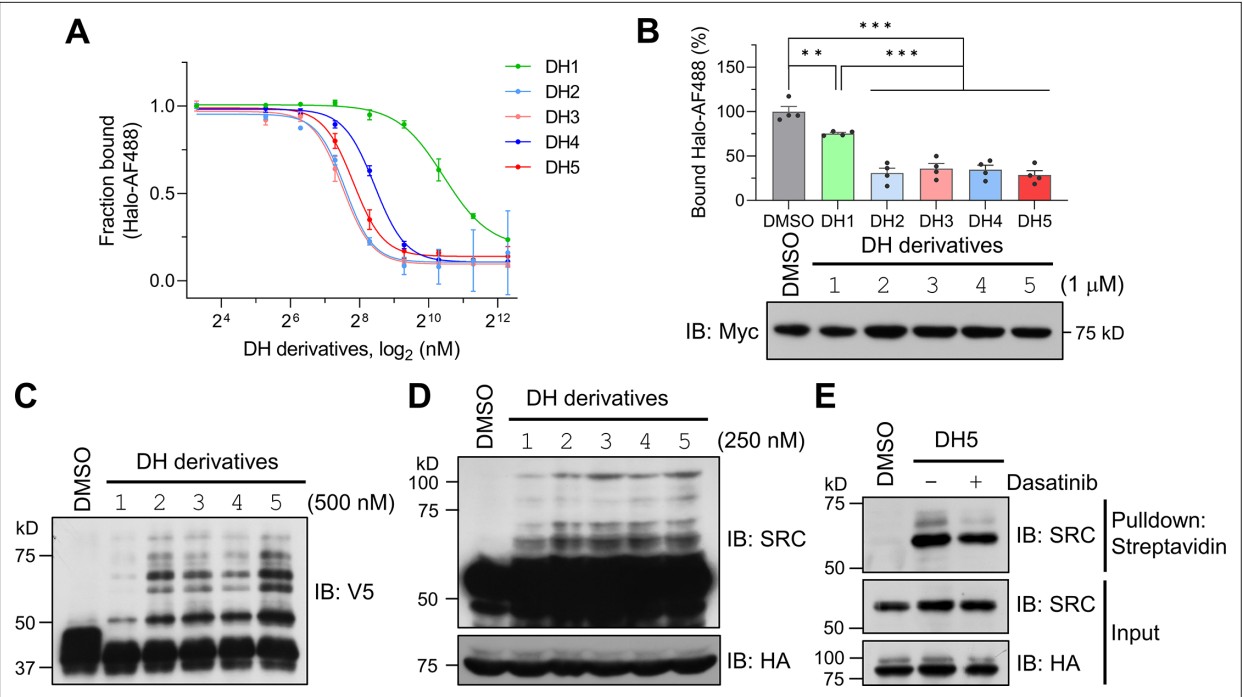

**Figure 4.** Dasatinib-HTL (DH) derivatives with modified and longer linkers enhances POST-IT performance. (**A**) In vitro binding assay for DH derivatives via fluorescence polarization (FP) assay. Purified Myc-Halo-PafA (15 nM) and 1 nM Halo-AF488 were incubated with various DH derivatives in a serial dilution at 37°C for 30 min prior to FP measurement. n = 2. Data are shown as mean ± s.d. (**B**) Cellular binding assay for DH derivatives by FP measurement. HEK293T cells transfected with Myc-Halo-PafA were later incubated with 1 µM of each DH derivative or DMSO for 3 hr, followed by cell lysate incubation with 2 nM of Halo-AF488 at 37°C for 30 min. n = 4. Data are shown as mean ± s.e.m. p-Values were calculated by an unpaired two-sided *t*-test. *p<0.05; **p<0.01; ***p<0.001; versus DMSO. Immunoblot (IB) presents a representative image of the input levels for each condition, demonstrating that treatment with DH derivatives did not alter the expression levels of Myc-Halo-PafA. (**C**) In vitro labeling activity comparison among DH derivatives. Purified recombinant Halo$^{8KR}$-PafA (0.5 µM), SBP$^{K4R}$-sPup$^{K61R}$ (10 µM), and SRC(247-536)-2xV5 (0.5 µM) were incubated with 0.5 µM of each DH derivative at 37°C for 30 min. (**D, E**) In cellular labeling activity comparison among DH derivatives. HA-Halo$^{8KR}$-PafA and SBP$^{K4R}$-sPup$^{K61R}$ were used for co-transfection, and cells were treated with 250 nM of various DH derivatives. Pupylation levels of purified short SRC (**C**), endogenous SRC (**D**), or pull-downed endogenous SRC (**E**) were assessed by IB using anti-V5 or anti-SRC antibodies for exogenous or endogenous SRC, respectively.

The online version of this article includes the following source data and figure supplement(s) for figure 4:

**Source data 1.** File containing labeled original western blots for *Figure 4*.

**Source data 2.** Original files for western blots displayed in *Figure 4*.

**Figure supplement 1.** Chemical structures of dasatinib and its HTL derivatives with varied linkers.

**Figure supplement 2.** Synthesis of dasatinib-Halo derivatives 2–5 (DH2-5).

**Figure supplement 3.** Representative spectra of DH2.

**Figure supplement 4.** Representative spectra of DH3.

**Figure supplement 5.** Representative spectra of DH4.

**Figure supplement 6.** Representative spectra of DH5.

**Figure supplement 7.** DH5 maintained robust kinase inhibitory activity.

SRC kinase activity and confirmed that DH5 retains comparable activity to dasatinib (*Figure 4—figure supplement 7*).

## Identification of target proteins for dasatinib by POST-IT

With our optimized POST-IT system and DH5, POST-IT$^{DH5}$, in hand, we conducted labeling experiments in live cells with or without an excess of dasatinib. After pulling down cell lysates with streptavidin beads, we analyzed biotin-eluted proteins via mass spectrometry. Remarkably, we identified eight known target proteins (FYN, RIPK2, MAPK14, ABL2, ABL1, CSK, LYN, and SRC) with significant enrichment and two known target proteins (GAK and SIK2) with lower confidence (*Figure 5A*, *Figure 5—source data 1*). Among these 10 known targets, 5 are tyrosine kinases

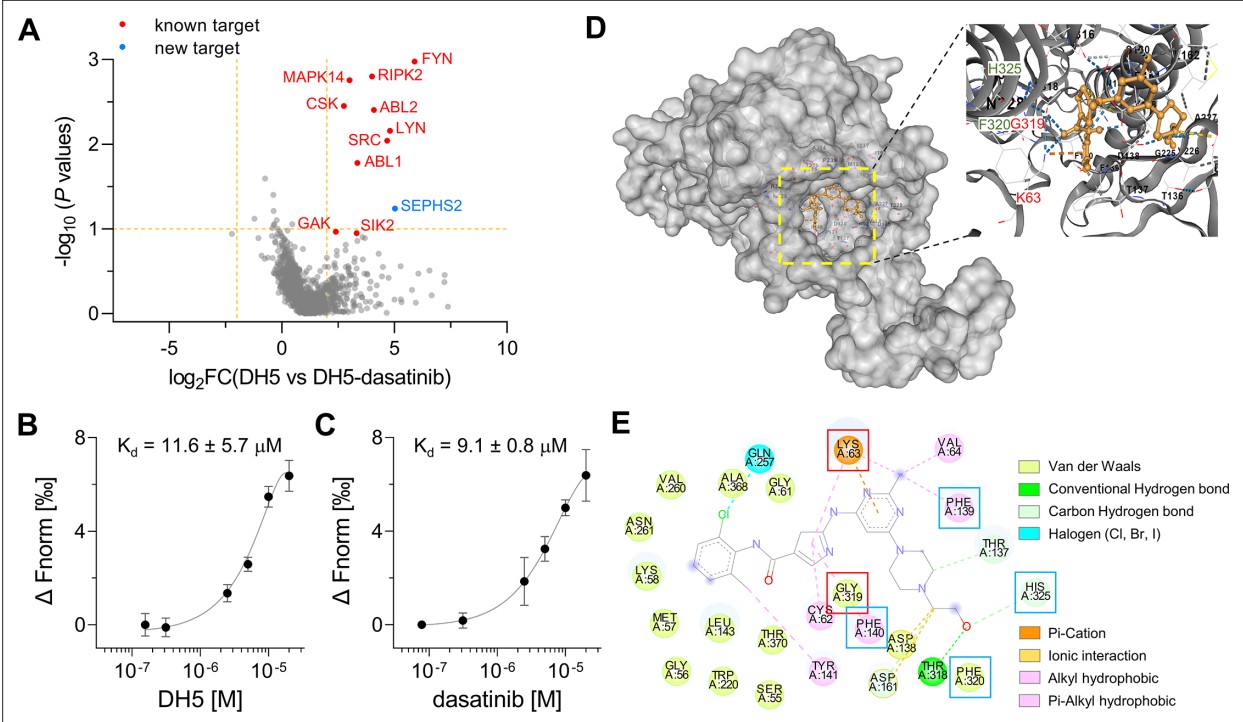

**Figure 5.** Target-ID by POST-IT$^{DH5}$. (**A**) Volcano plot displaying the relative fold change (FC) of DH5-binding proteins compared to those with dasatinib competition. Known target proteins for dasatinib were highlighted in red, and a newly identified target protein, SEPHS2, was marked in blue. (**B, C**) Microscale thermophoresis (MST) analysis demonstrates a direct interaction between DH5 and Cy5-labeled SEPHS2 (**B**) or between dasatinib and Cy5-labeled SEPHS2 (**C**), yielding $K_d$ values of 11.6 ± 5.7 μM and 9.1 ± 0.8 μM, respectively. n = 2. Data are shown as mean ± s.d. (**D**) Molecular docking binding pose between dasatinib and SEPHS2. The yellow box indicates the binding pocket, expanded for a closer view. The binding pocket of dasatinib precisely fits into the active sites of SEPHS2. (**E**) Diagram of 2D molecular docking interaction between SEPHS2 and dasatinib. Red boxes highlight the active sites, and blue boxes indicate residues among HUB nodes.

The online version of this article includes the following source data for figure 5:

**Source data 1.** Proteomics result for DH5.

(FYN, ABL1/2, CSK, LYN, and SRC) and 4 are serine/threonine kinases (RIPK2, MAPK14, GAK, and SIK2). These results validate the effectiveness of our POST-IT system as a target-ID methodology and underscore its potential superiority to other PL systems for target-ID (*Hill et al., 2016*; *Kwak et al., 2022*).

Additionally, we identified selenophosphate synthetase 2 (SEPHS2) as a novel protein target that binds to dasatinib. SEPHS2, a crucial enzyme for selenoprotein synthesis, exhibits ATP-binding and kinase activities, supporting its potential interaction with dasatinib. To confirm SEPHS2 as a genuine target of dasatinib, we purified the recombinant SEPHS2 protein and measured its binding affinity to DH5 using microscale thermophoresis (MST) analysis (*Figure 5B and C*). This analysis yielded a dissociation constant ($K_d$) of 11.6 ± 5.7 μM for DH5 and 9.1 ± 0.8 μM for dasatinib. To elucidate the binding mode of dasatinib to SEPHS2, we performed molecular docking analysis to model their interaction. A previous study using molecular dynamics simulations identified Sec60, Lys63, and Gly319 as active sites and 15 amino acids as key residues or HUB nodes (*Nunziata et al., 2019*). Remarkably, the docking analysis revealed a favorable binding mode for dasatinib within the active sites of SEPHS2, with a Vina score of −9.9 (*Figure 5D*). In the binding pocket of SEPHS2, dasatinib engages in multiple interactions, including pi-cation or cation-π interaction with Lys63 and Van der Waals interaction with Gly319 in the active sites, along with hydrophobic interactions with Phe130 and Phe140, Van der Waals interaction with Phe320, and a hydrogen bond with His325 among the HUB nodes (*Figure 5E*). Collectively, these results suggest that SEPHS2 is a novel protein target for dasatinib, indicating that dasatinib may impact SEPHS2 function.

## Identification of target proteins for chloroquine by POST-IT

To further demonstrate the effectiveness of the POST-IT system in target-ID, we applied it to identify target proteins for HCQ, using it as an additional model ligand. HCQ has a long history of clinical use for treating malaria, rheumatoid arthritis, and lupus. Moreover, it has been widely used in research as an autophagy inhibitor, a property gaining interest in the development of cancer therapy (*Jain et al., 2023*). Despite its broad application, the precise mechanisms through which HCQ inhibits autophagy and the pathways through which HCQ or chloroquine (CQ) elicit adverse effects such as psychiatric symptoms, retinal and ototoxicity, cardiac toxicity, and even death (*Muller, 2021*; *Nirk et al., 2020*) remain to be fully understood.

First, we synthesized a dimer form, DC661-H1, of a CQ HTL derivative with the same linker used for DH5 (*Figure 6—figure supplements 1A*; *Figure 6—figure supplement 2* and *Figure 6—figure supplement 3*), assuming that a dimer would enhance binding affinity as previously described (*Rebecca et al., 2019*). We confirmed that DC661-H1 and DC660, an analog of DC661-H1 without HTL, induced robust LC3-II bands in two mammalian cell lines to a similar extent as HCQ, indicating that they function as potent autophagy inhibitors (*Figure 6—figure supplement 1B*). To test the versatility of POST-IT as a tool for mass analysis, POST-IT was coupled with stable isotope labeling with amino acid in cell culture (SILAC) for target-ID. Heavy-Lys/Arg-labeled HEK293T cells expressing POST-IT were treated with DC661-H1 and DMSO, while light-Lys/Arg-labeled HEK293T cells expressing POST-IT were treated with DC661-H1 and an excess of competitive DC660, as described in *Figure 6—figure supplement 4*. Intriguingly, POST-IT identified several target proteins directly linked to autophagy regulation (*Figure 6A*, *Figure 6—source data 3*). Hence, we aimed to verify whether DC660 and HCQ bind to these autophagy-related proteins. Among them, we purified five recombinant proteins, including TOM1, TOM1L2, TRAPPC3, VPS29, and VPS37C, and measured their binding properties using MST analysis. TOM1, TOM1L2, and TRAPPC3 appeared to bind well to DC660 but failed to acquire $K_d$ values due to abnormal thermophoretic movement, suggesting non-homogeneous aggregation (*Figure 6—figure supplement 5A–C*). VPS29 showed dose-dependent binding to DC660, yielding a $K_d$ of 24.95 ± 11.71 µM (*Figure 6—figure supplement 5D*). However, binding to HCQ was negligible in these proteins (*Figure 6—figure supplement 5E–H*). These results support the validity of our POST-IT as a PL system for live cell target-ID, as all tested proteins showed binding ability to DC660.

Next, we investigated the binding affinity of VPS37C to DC660 and HCQ using purified recombinant protein. MST analysis revealed a very high binding affinity of VPS37C to DC660 with a $K_d$ value of 21.5 ± 9.8 nM (*Figure 6B*). Notably, VPS37C also exhibited moderate binding affinity to HCQ (*Figure 6C*). This result prompted us to further explore the binding ability of VPS37C under cellular conditions. First, we performed POST-IT labeling in HEK293T cells expressing VPS37C with a V5-tag, followed by pulldown using streptavidin beads. VPS37C was substantially enriched in the presence of DC661-H1, but its binding was almost completely abolished by competitive DC660 or HCQ (*Figure 6D and E*). Furthermore, through a CETSA experiment, we observed that VPS37C became more stable in the presence of DC660 or HCQ than in the DMSO condition, upon exposure to increasing temperatures from cell lysates (*Figure 6F and G*). Collectively, these results provide strong evidence that VPS37C binds to DC661 (or DC660) as well as HCQ in vitro and in live cells, suggesting a potential role of VPS37C in the action mechanism of CQ or HCQ in inhibiting autophagy.

## In vivo application of POST-IT as a target-ID system

Phenotype-based screening is gaining increased recognition as a powerful strategy for discovering novel compounds, superior drugs, or uncovering unknown biological mechanisms, such as the non-canonical translation discovered in our previous study using this approach (*Jin et al., 2018*). Zebrafish have emerged as an excellent in vivo whole-animal model system for small molecule screening and drug discovery. They are vertebrates, highly prolific, and amenable to high-throughput chemical screens, often leading to the serendipitous discovery of novel compounds (*Patton et al., 2021*). However, target-ID remains a major hurdle in understanding the mechanisms of action of novel compounds, thereby delaying the advancements in drug development. Therefore, we set out to test the applicability of POST-IT as an efficient method for target-ID in zebrafish.

To this end, we first generated a robust, ubiquitous expression plasmid for POST-IT in zebrafish using an optimized QF-binary system (*Burgess et al., 2020*). This system combines a DNA binding

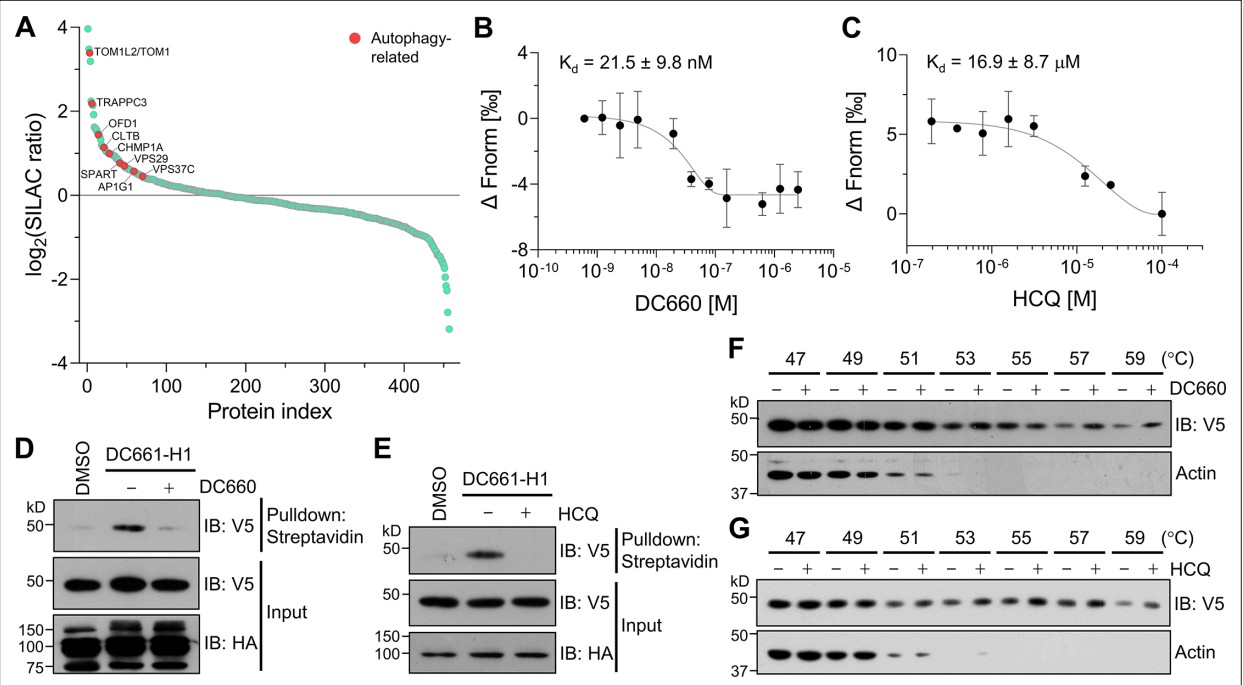

**Figure 6.** Target-ID for hydroxychloroquine (HCQ) by POST-IT$^{DC661-H1}$. (**A**) Rank plot analysis of SILAC ratio values of proteins in three biological replicates. Autophage-related proteins ranked highly are marked in red. Heavy-labeled cells were treated with 200 nM DC661-H1, whereas light-labeled cells underwent incubation with both DC661-H1 (200 nM) and competitive DC660 (2 µM). (**B, C**) Microscale thermophoresis (MST) analyses demonstrate direct interactions of DC660 (**B**) and HCQ (**C**) with VPS37C, yielding $K_d$ values of 21.5 ± 9.8 nM and 16.9 ± 8.7 µM, respectively. n = 2. Data are shown as mean ± s.d. (**D, E**) Immunoblot results indicate that the VPS37C-V5 protein from cells treated with DC661-H1 was significantly enriched after streptavidin pulldown. The competition between DC661-H1 and either DC660 (**D**) or HCQ (**E**) nearly completely abolished VPS37C binding. HEK293T cells were co-transfected with HA-Halo$^{8KR}$-PafA$^{S126A,K172R}$, SBP$^{K4R}$-sPup$^{K61R}$, and VPS37C-V5. After 24 hr, cells were incubated with 200 nM of DC661-H1, with or without 2 µM of DC660 (**D**) or HCQ (**E**). Twenty-four hours later, cells were collected for further analysis. (**F, G**) Cellular thermal shift assay (CETSA) results demonstrate that VPS37C becomes thermostable when exposed to DC660 (**F**) or HCQ (**G**). Selective stabilization of VPS37C is evident at temperatures of 53°C or higher, a phenomenon not observed in β-actin.

The online version of this article includes the following source data and figure supplement(s) for figure 6:

**Source data 1.** File containing labeled original western blots for *Figure 6*.

**Source data 2.** Original files for western blots displayed in *Figure 6*.

**Source data 3.** SILAC result for DC661-H1.

**Figure supplement 1.** A DC661-HTL derivative, DC661-H1, is highly active as an autophagy inhibitor.

**Figure supplement 1—source data 1.** File containing labeled original western blots for *Figure 6—figure supplement 1*.

**Figure supplement 1—source data 2.** Original files for western blots displayed in *Figure 6—figure supplement 1*.

**Figure supplement 2.** Synthesis of DC661-Halo 1 (DC661-H1).

**Figure supplement 3.** Representative spectra of DC660.

**Figure supplement 4.** Schematic of the experimental flow for applying POST-IT in SILAC.

**Figure supplement 5.** Microscale thermophoresis (MST) binding analysis of candidate proteins identified with DC661-H1.

**Figure supplement 6.** Molecular docking binding poses of DC661 and HCQ with the VPS37 family.

domain from the QF transactivator fused with an activation domain from the Gal4 transactivator, creating QFGal4, which binds to five repeats of QUAS, leading to gene activation. POST-IT, tagged with a 3×FLAG, is inserted following a 2A ribosome-skipping sequence, allowing for the simultaneous expression of QFGal4 and POST-IT under the control of a ubiquitous promoter, *ubb* (**Figure 7A**). We chose the *ubb* promoter over tissue-specific promoters to more accurately evaluate the levels and potential toxicity of transgene expression. The expression of SBP$^{K4R}$-sPup$^{K61R}$ is driven by a 5×QUAS promoter, and EGFP is incorporated upstream of a 2A sequence as an expression marker. All these components are assembled in a single plasmid to simplify transgenesis. Embryos injected with this

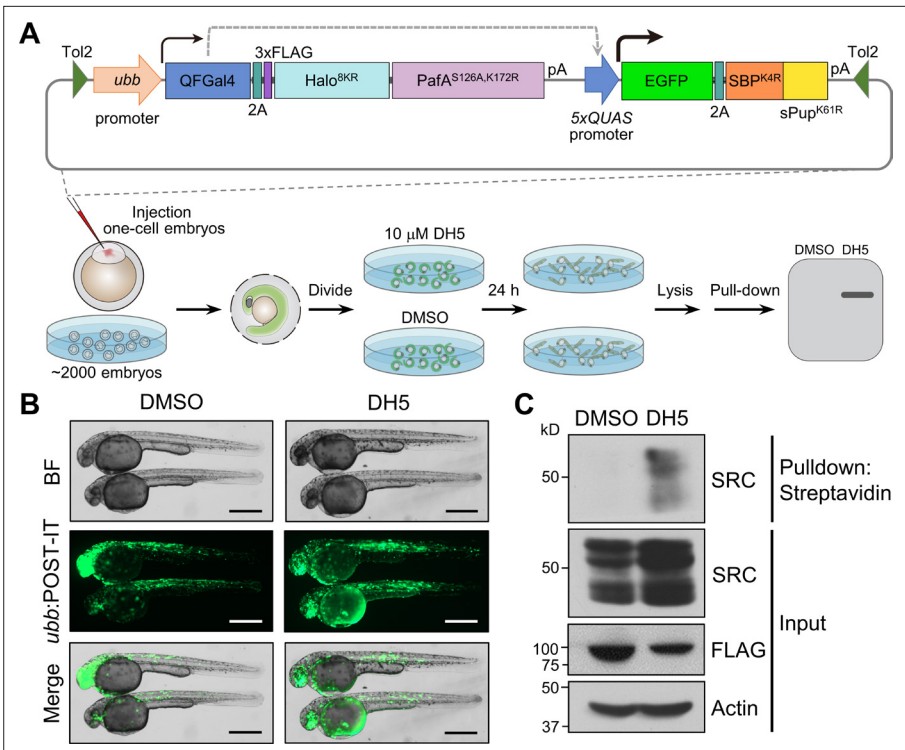

**Figure 7.** POST-IT enables in vivo target-ID. (**A**) Schematic of experimental design for applying POST-IT in zebrafish. A plasmid construct containing the POST-IT system was injected into embryos at the one-cell stage. The injected embryos were dechorionated 24 hr later, and then treated with DH5 or DMSO for an additional 24 hr. Embryos lysates were collected and analyzed by immunoblot. (**B**) Representative images of injected embryos. Bright-field (BF) images reveal no overt toxicity from POST-IT expression. EGFP images display robust expression of POST-IT. Scale bar, 500 μm. (**C**) Immunoblot analysis demonstrates that POST-IT exhibits significant enrichment of SRC in the presence of DH5 but not DMSO.

The online version of this article includes the following source data and figure supplement(s) for figure 7:

**Source data 1.** File containing labeled original western blots for *Figure 7*.

**Source data 2.** Original files for western blots displayed in *Figure 7*.

**Figure supplement 1.** Expression of POST-IT does not cause toxicity in zebrafish.

plasmid displayed robust EGFP expression throughout their entire bodies without observable toxicity, indicating the suitability of the POST-IT system in zebrafish (*Figure 7B*, *Figure 7—figure supplement 1A*). In addition, we generated a transgenic zebrafish line that expresses POST-IT under a heat-shock promoter. Embryos from this transgenic line show no obvious toxicity after heat shock (*Figure 7—figure supplement 1B and C*). Note that the expression pattern of EGFP is non-homogeneous due to the mosaic expression of the injected plasmid (*Figure 7B*). After treating live embryos with DH5 or DMSO, we observed marked enrichment of endogenous SRC in embryos treated with DH5 but not in those treated with DMSO, although expression levels of POST-IT were slightly lower in the DH5 group due to injection variability (*Figure 7C*). Together, these results demonstrated that POST-IT can be effectively utilized as a transgene in zebrafish without toxicity and applied for in vivo target-ID.

## Discussion

In this study, we present POST-IT, a new non-diffusive PL system for target-ID in live cells and in vivo animal models. We systematically employed mutagenesis to eliminate self-pupylation, polypupyla-tion, and depupylase activity. Moreover, we optimized the linker length between HaloTag and PafA and developed PEG linkers for HTL derivatives. POST-IT provides a straightforward, yet highly robust and reliable method for target-ID within intact cellular context, a feature that significantly reduces artifacts from cell lysates and may be crucial for certain ligand/drug–protein interactions. We further

demonstrated the applicability of POST-IT in an animal model, zebrafish. POST-IT does not cause noticeable toxicity and allows for tissue-specific and temporally controlled expression using a variety of transgenic zebrafish systems. We believe that POST-IT can be readily applied to other organisms, from *C. elegans* to mice, while retaining its effectiveness. This is evidenced by the robust activity of Halo-PafA, which remains active even at temperatures as low as 20°C (*Figure 1—figure supplement 1B*).

While TurboID is a widely used tool for studying protein networks, it suffers from persistent high background labeling, even without biotin. This necessitates meticulous experimental adjustments, such as employing dialyzed FBS and fine-tuning biotin labeling levels (*May et al., 2020*). Alternative strategies, such as light-regulation (*Lee et al., 2023*) and split-TurboID (*Cho et al., 2020*), have been introduced to mitigate high background labeling, yet their suitability for target-ID and in vivo applications remains to be explored. Additionally, previous studies have reported cellular toxicity associated with TurboID (*Branon et al., 2018*; *May et al., 2020*), although recent work shows no obvious toxicity in zebrafish (*Xiong et al., 2021*). Furthermore, the labeling radius of TurboID can exceed 35 nm (*May et al., 2020*), complicating the precise identification of target proteins. While these PL systems can provide valuable insights into a drug's protein network (*Kwak et al., 2022*; *Tao et al., 2023*), direct target-ID is essential for understanding the drug's mechanism of action.

A recent study introduced BmTyr, a PL system based on *Bacillus megaterium* tyrosinase (*Zhu et al., 2024*), which oxidizes phenol or catechol derivatives into *o*-quinones for labeling adjacent proteins. This method provides fast and lower-background labeling compared to TurboID. Unlike APEX, BmTyr does not require hydrogen peroxide, enhancing its biocompatibility for in vivo use. However, the need for copper might introduce toxicity and limitations. Oxygen dependency and glutathione reactivity may also affect its performance. While in vivo application was demonstrated by injecting purified BmTyr into mouse brains, the feasibility of its transgenic expression and broader application remains untested. Importantly, the diffusive nature of BmTyr's labeling might challenge its use for precise target-ID, despite being an innovative alternative PL toolkit.

Using our POST-IT system, we identified SEPHS2 and VPS37C as novel target proteins for dasatinib and HCQ/CQ, respectively. SEPHS2 detoxifies selenide, a toxic intermediate of selenocysteine biosynthesis, by converting it with ATP into selenophosphate, an essential component for selenoprotein production. This function is particularly important in cancer cells that require enhanced defense against oxidative stress. These cancer cells frequently exhibit elevated SLC7A11/xCT expression, leading to increased selenide import and a heightened demand for selenide detoxification. Similarly, abnormal upregulation of selenium uptake and selenocysteine biosynthesis occurs in many cancer cells, including breast cancers (*Carlisle et al., 2020*) and acute myeloid leukemia (*Eagle et al., 2022*), making SEPHS2 crucial for their survival but not for normal cells. Therefore, SEPHS2 is a promising target for cancer therapy. Given that SEPHS2 has been identified as a target for dasatinib, potentially binding to the active site, investigating whether SEPHS2 inhibition by dasatinib contributes to its anticancer effects is warranted. Binding information of dasatinib to SEPHS2 could facilitate the development of new SEPHS2 inhibitors.

Despite decades of research using HCQ/CQ as an autophagy inhibitor, its exact mechanism of action remains elusive. Although it has been widely assumed that HCQ/CQ increases lysosomal pH, *Mauthe et al., 2018* demonstrated that its primary effect is to inhibit the fusion between autophagosomes and lysosomes. Additionally, HCQ/CQ induces disorganization in the Golgi and endosomal systems independently of autophagy. These findings highlight the significance of pH-independent mechanisms in HCQ/CQ's action. Furthermore, components of the Endosomal Sorting Complex Required for Transport (ESCRT) complexes have been shown to regulate the final stages of autophagosome formation (*Javed et al., 2023*; *Takahashi et al., 2019*; *Takahashi et al., 2018*; *Zhen et al., 2020*). ESCRT-I consists of three core components (TSG101/VPS23, VPS28, and one of VPS37A/B/C/D) and a single auxiliary protein among UBAP1, MVB12A, and MVB12B. While VPS37A has been identified as a key player in phagophore closure (*Javed et al., 2023*; *Takahashi et al., 2019*), the roles of other VPS37 proteins remain unexplored. Considering the structural similarities among the VPS37 proteins (*Figure 6—figure supplement 6*), it is plausible that HCQ/CQ could interact with multiple members of this family to some extent, potentially inhibiting their role in autophagosome maturation. This suggests two potential mechanisms for HCQ/CQ-mediated autophagy inhibition. One involves interfering with the function of VPS37 proteins in autophagosome closure. The other suggests that

HCQ/CQ is actively transported into lysosomes by ESCRT-I, leading to lysosomal alkalization. Further research is crucial to elucidate these mechanisms. Understanding these pathways will not only illuminate therapeutic strategies for HCQ/CQ but also guide the development of superior derivatives with minimized side effects.

There remain some limitations in utilizing POST-IT system. One limitation is the requirement to synthesize an HTL derivative for the compound of interest, a challenge common to all target-ID methods except for those that are modification-free. To facilitate derivatization, we provide guidelines for linkers incorporating carbamate and PEG, as shown in *Figure 4* and *Figure 4—figure supplement 1*. PEG linkers are advantageous for their increased solubility and cell permeability. Provided the derivatives maintain comparable binding affinity to the target protein or preserve the compound's activity, we believe that POST-IT can be effectively applied from in vitro conditions to live cells and organisms. Secondly, POST-IT may require significantly longer incubation times in live cells compared to in vitro reactions, which could be attributed to several factors, such as insufficient expression of POST-IT or Pup, or the instability or metabolism of an HTL derivative of the drug. One potential approach to improve reproducibility is to generate stable cell lines and transgenic zebrafish lines that express high levels of POST-IT, which is feasible due to the lack of observable toxicity from POST-IT expression. High levels of Pup expression are especially important for efficient tagging. Establishing stable cell lines with robust POST-IT expression could also enhance scalability for drug discovery applications. Additionally, further improvements of POST-IT through directed evolution, rational protein engineering, and structure-aided design are warranted. Studying the metabolism, stability, and toxicity of HTL derivatives of the drug is also valuable for better experimental design. Given that many drugs function in specific cellular locations, such as the cell surface and mitochondria, targeting POST-IT to precise subcellular locations could enhance the likelihood of identifying true biological target proteins rather than off-target binding partners. Therefore, exploring the functionalities of POST-IT in specific subcellular contexts is an essential direction for future research. Additionally, in this study, we utilized HEK293T cells for proof of concept. However, investigating protein targets in diverse cell types, particularly cancer cells, may reveal a broader spectrum of targets. To gain a more comprehensive understanding of a drug's target profile, we recommend that users employ POST-IT in various cell lines or in vivo models, such as zebrafish.

In conclusion, POST-IT represents a significant advancement in the field of target identification, offering an innovative, non-diffusive proximity tagging system designed for precise target-ID within live cells and in vivo models. By preserving the cellular context, POST-IT overcomes the limitations associated with traditional cell lysate methods and offers superior specificity compared to existing PL systems like TurboID. We successfully identified SEPHS2 as a target for dasatinib and VPS37C as a target for HCQ, exemplifying its effectiveness. Its application in both live cells and zebrafish embryos underscores its versatility and potential to provide valuable insights into the molecular mechanisms of drug action, thereby enhancing the scope of biomedical research and therapeutic development.

# Materials and methods

**Key resources table**

| Reagent type (species) or resource | Designation | Source or reference | Identifiers | Additional information |
|---|---|---|---|---|
| Gene (*Homo sapiens*) | SRC | NCBI | NM_005417.5 | |
| Gene (*H. sapiens*) | VPS37C | NCBI | NM_017966.5 | |
| Gene (*H. sapiens*) | SEPHS2 | NCBI | NM_012248.4 | |
| Strain, strain background (*Danio rerio*, AB) | Zebrafish | Institute of Hydrobiology, Wuhan, China | | |
| Strain, strain background (*Escherichia coli*) | BL21(DE3) | Weidibio | Cat# EC1002 | Electrocompetent cells |
| Strain, strain background (*E. coli*) | ArticExpress(DE3) | HaiGene | Cat# K0004 | Electrocompetent cells |

*Continued on next page*

*Continued*

| Reagent type (species) or resource | Designation | Source or reference | Identifiers | Additional information |
|---|---|---|---|---|
| Cell line (*H. sapiens*) | HEK293T | ATCC | CRL-3216, RRID:CVCL_0063 | |
| Cell line (*H. sapiens*) | A375 | ATCC | CRL-1619, RRID:CVCL_A375 | |
| Antibody | Anti-LC3B (rabbit monoclonal) | ABclonal | Cat# A19665, RRID:AB_2862723 | WB (1:10,000) |
| Antibody | Anti-V5-tag (rabbit monoclonal) | ABclonal | Cat# AE089 | WB (1:5000) |
| Antibody | Anti-Myc-tag (rabbit monoclonal) | ABclonal | Cat# AE070, RRID:AB_2863795 | WB (1:5000) |
| Antibody | Anti-HA-tag (mouse monoclonal) | ABclonal | Cat# AE008, RRID:AB_2770404 | WB (1:2500) |
| Antibody | Anti-Flag-tag (rabbit monoclonal) | ABclonal | Cat# AE092, RRID:AB_2940847 | WB (1:10,000) |
| Antibody | Anti-β-Actin (rabbit monoclonal) | ABclonal | Cat# AC026, RRID:AB_2768234 | WB (1:10,000) |
| Antibody | Anti-Src(36D10) (rabbit monoclonal) | Cell Signaling Technology | Cat# 2109, RRID:AB_2106059 | WB (1:5000) |
| Recombinant DNA reagent | pET28a-Myc-Halo-PafA | This paper | | Plasmid construction (Dr. YN Jin lab) |
| Recombinant DNA reagent | pET28a-HB-Pup | This paper | | Plasmid construction (Dr. YN Jin lab) |
| Recombinant DNA reagent | pET28a-SBP-sPup | This paper | | Plasmid construction (Dr. YN Jin lab) |
| Recombinant DNA reagent | pET28a-TS-sPup | This paper | | Plasmid construction (Dr. YN Jin lab) |
| Recombinant DNA reagent | pET28a-TS-sPup(K61R) | This paper | | Plasmid construction (Dr. YN Jin lab) |
| Recombinant DNA reagent | pET28a-TS(K1,2R)-sPup(K61R) | This paper | | Plasmid construction (Dr. YN Jin lab) |
| Recombinant DNA reagent | pET28a-SBP(K4R)-sPup(K61R) | This paper | | Plasmid construction (Dr. YN Jin lab) |
| Recombinant DNA reagent | pET28a-sSBP-sPup(K61R) | This paper | | Plasmid construction (Dr. YN Jin lab) |
| Recombinant DNA reagent | pET28a-Halo(8KR)-PafA-6xHis | This paper | | Plasmid construction (Dr. YN Jin lab) |
| Recombinant DNA reagent | pET28a-mycHalo-PafA(S126A) | This paper | | Plasmid construction (Dr. YN Jin lab) |
| Recombinant DNA reagent | pGEX-6P-2-PafA | This paper | | Plasmid construction (Dr. YN Jin lab) |
| Recombinant DNA reagent | pGEX-6P-2-2xmycPafA(S126A)-Halo(8KR) | This paper | | Plasmid construction (Dr. YN Jin lab) |
| Recombinant DNA reagent | pET28a-SRC(251-536)-2xV5 | This paper | | Plasmid construction (Dr. YN Jin lab) |
| Recombinant DNA reagent | pET28a-N-SBPII-SEPHS2(U60C)-2xV5 | This paper | | Plasmid construction (Dr. YN Jin lab) |
| Recombinant DNA reagent | pET28a-SAVSBPM18-XTEN-DBD | This paper | | Plasmid construction (Dr. YN Jin lab) |

*Continued on next page*

*Continued*

| Reagent type (species) or resource | Designation | Source or reference | Identifiers | Additional information |
|---|---|---|---|---|
| recombinant DNA reagent | pTol2-3xHA-FKBP-EGFP-pA | This paper | | Plasmid construction (Dr. YN Jin lab) |
| Recombinant DNA reagent | pTol2-3xV5-FRB-mKate2-PafA-2ApA-CK2 | This paper | | Plasmid construction (Dr. YN Jin lab) |
| Recombinant DNA reagent | pEF6a~HB-PupE | PMID:30104635 | | Plasmid construction (Dr. YN Jin lab) |
| Recombinant DNA reagent | pEF6a~SBP-sPup | This paper | | Plasmid construction (Dr. YN Jin lab) |
| Recombinant DNA reagent | pEF6a~SBP(K4R)-sPup(K61R) | This paper | | Plasmid construction (Dr. YN Jin lab) |
| Recombinant DNA reagent | pEF6a~sSBP-sPup(K61R) | This paper | | Plasmid construction (Dr. YN Jin lab) |
| Recombinant DNA reagent | pTol2-HA-Halo-PafA-2ApA-CK2 | This paper | | Plasmid construction (Dr. YN Jin lab) |
| Recombinant DNA reagent | pTol2-HA-His-Halo(8KR)-5aa-PafA-His | This paper | | Plasmid construction (Dr. YN Jin lab) |
| Recombinant DNA reagent | pTol2-HA-His-Halo(8KR)-10aa-PafA-His | This paper | | Plasmid construction (Dr. YN Jin lab) |
| Recombinant DNA reagent | pTol2-HA-His-Halo(8KR)-18aa-PafA-His | This paper | | Plasmid construction (Dr. YN Jin lab) |
| Recombinant DNA reagent | pTol2-HA-His-Halo(8KR)-40aa-PafA-His | This paper | | Plasmid construction (Dr. YN Jin lab) |
| Recombinant DNA reagent | pTol2-HA-His-Halo(8KR)-18aa-PafA(S126A)-His | This paper | | Plasmid construction (Dr. YN Jin lab) |
| Recombinant DNA reagent | pTol2-HA-His-Halo(8KR)-18aa-PafA(S126A,K172R)-His | This paper | | Plasmid construction (Dr. YN Jin lab) |
| Recombinant DNA reagent | pTol2-SRC(247-536)-2xV5-2ApA-CK2 | This paper | | Plasmid construction (Dr. YN Jin lab) |
| Recombinant DNA reagent | pEF6a~SBP(K4R)-tbsPup(K61R) | This paper | | Plasmid construction (Dr. YN Jin lab) |
| Recombinant DNA reagent | pTol2-HA-His-Halo(8KR)-tbPafA-His | This paper | | Plasmid construction (Dr. YN Jin lab) |
| Recombinant DNA reagent | pTol2-HA-His-tbPafA-Halo(8KR)-His | This paper | | Plasmid construction (Dr. YN Jin lab) |
| Recombinant DNA reagent | PB-Tet-Halo(8KR)-PafA(S126A,K172R)-Ter3G-SBP(K4R)-sPup(K61R)-ires-PuroR-P2A-EGFP | This paper | | Plasmid construction (Dr. YN Jin lab) |
| Recombinant DNA reagent | pHAGE-ires-Puro-VPS37C-2xV5 | This paper | | Plasmid construction (Dr. YN Jin lab) |
| Recombinant DNA reagent | pUbi-QFGal4-2A-POST-IT | This paper | | Plasmid construction (Dr. YN Jin lab) |
| Recombinant DNA reagent | pHsp-QFGal4-2A-POST-IT | This paper | | Plasmid construction (Dr. YN Jin lab) |
| Peptide, recombinant protein | RR-SRC peptide (RRLIEDAEYAARG) | Sangon | Cat# T510264-0001 | |
| Commercial assay or kit | Kinase-Lumi Luminescent Kinase Assay Kit | Beyotime | Cat# S0150S | |
| Chemical compound, drug | DH1 | This paper | | Chemical synthesis (Dr. H-B Zhou lab) |

*Continued on next page*

*Continued*

| Reagent type (species) or resource | Designation | Source or reference | Identifiers | Additional information |
|---|---|---|---|---|
| Chemical compound, drug | DH2 | This paper | | Chemical synthesis (Dr. H-B Zhou lab) |
| Chemical compound, drug | DH3 | This paper | | Chemical synthesis (Dr. H-B Zhou lab) |
| Chemical compound, drug | DH4 | This paper | | Chemical synthesis (Dr. H-B Zhou lab) |
| Chemical compound, drug | DH5 | This paper | | Chemical synthesis (Dr. H-B Zhou lab) |
| Chemical compound, drug | DC661-H1 | This paper | | Chemical synthesis (Dr. H-B Zhou lab) |
| Chemical compound, drug | DC660 | This paper | | Chemical synthesis (Dr. H-B Zhou lab) |
| Chemical compound, drug | Hydroxychloroquine | TargetMol | Cat# T9287 | |
| Chemical compound, drug | Dasatinib | Shanghai yuanye Bio-Technology | Cat# S45672-25MG | |
| Chemical compound, drug | HaloTag Alexa Fluor 488 Ligand | Promega | Cat# G1001 | |
| Chemical compound, drug | HaloTag Biotin Ligand | Promega | Cat# G8281 | |
| Software, algorithm | Fiji | PMID:22743772 | RRID:SCR_002285 | |
| Software, algorithm | Prism 9.5 | GraphPad Software | RRID:SCR_002798 | |
| Software, algorithm | MaxQuant (2.1.0.0) | PMID:19029910 | RRID:SCR_14485 | |
| Software, algorithm | FragPipe software v20.0 | PMID:28394336 | RRID:SCR_14485 | |
| Other | Streptavindin-HRP | Cell Signaling Technology | Cat# 3999, RRID:AB_10830897 | WB (1:5000), for Halo-biotin competition assay |
| Other | Dialyzed fetal bovine serum (FBS) | VivaCell | Cat# C3820-0100 | For SILAC experiment |
| Other | L-Lysine·HCl | aladdin | Cat# L113006-25g | For SILAC experiment |
| Other | L-Arginine·HCl | aladdin | Cat# 1119-34-25g | For SILAC experiment |
| Other | L-Proline | aladdin | Cat# P120032-25g | For SILAC experiment |
| Other | $^{13}C_6{}^{15}N_2$ labeled L-lysine·HCl | Silantes | Cat# 211603902 | For SILAC experiment |
| Other | $^{13}C_6{}^{15}N_4$ labeled L-arginine·HCl | Silantes | Cat# 201603902 | For SILAC experiment |
| Other | SILAC DMEM | Silantes | Cat# 280001200 | For SILAC experiment |

## Plasmid construction

Plasmids were typically constructed by the HiFi DNA assembly method using 2×MultiF Seamless Assembly Mix (RK21020, ABclonal, China). PCR amplification was carried out with a high-fidelity DNA polymerase, Phanta Max Super-Fidelity DNA Polymerase (P505-d1, Vazyme, China). All restriction enzymes were from New England Biolabs, USA. In general, the pTol2-EGFP-2ApA-CK2 or pTol2-EGFPpA vector, developed in our lab (*Jin et al., 2018*), was used for mammalian expression, and the pET28a vector was utilized for recombinant protein expression in *Escherichia coli*, unless specified otherwise. All DNA constructs were validated by Sanger sequencing.

To construct Halo-PafA plasmids, HaloTag and PafA were PCR amplified using HaloTag7 (a gift from Yang Yu, Chinese Academy of Sciences) and pEF6a-kozak-CD28-PafA-myc (a gift from Min Zhuang, ShanghaiTech University), respectively, and then subcloned into the NotI and EcoRI sites of pTol2-EGFP-2ApA-CK2, or into the EcoRI site of pET28a. HaloTag[8KR], TS-sPup, and SBP-sPup DNA fragments were synthesized by Tsingke Biotechnology (Wuhan). Halo[8KR]-PafA was subcloned into the NotI and EcoRI sites of pTol2-EGFP-pA or into the BamHI and NotI sites of pET28a. TS-sPup and

SBP-sPup were subcloned into the KpnI and NotI sites of pEF6a-HB-PUP (a gift from Min Zhuang) for mammalian expression and inserted into the EcoRI site of pET28a for *E. coli* expression. Point mutations, including SBP$^{K4R}$, TS$^{K8R}$, sPup$^{K61R}$, PafA$^{S126A}$, and PafA$^{K172R}$, were introduced using a modified site-directed mutagenesis (*Liu and Naismith, 2008*). For recombinant protein expression of PafA in *E. coli*, PCR-amplified PafA was fused to a GST-tag into the BamHI and EcoRI sites of pGEX-6P-2, generating pGEX-6P-2-PafA.

Target proteins, such as SRC, SEPHS2, VPS37C, TOM1, TOM1L2, TRAPPC3, and VPS29, were PCR amplified using a cDNA library synthesized with the HiScript III first-strand cDNA Synthesis Kit (R312, Vazyme, China) from total RNA extracted from HEK293T cells with RNAiso Plus (9109, Takara Bio, Japan). A catalytic active, truncated version of SRC spanning amino acids 247–536, SRC(247-536), was inserted into the NotI and EcoRI sites of pTol2-EGFP-2ApA-CK2, and into the BamHI and NotI sites of pET28a. SEPHS2-2xV5 was subcloned into the BamHI and EcoRI sites of pE28a-N-SBPII, engineered by inserting a SBPII-tag into pET28a vector in our lab for *E. coli* expression. The selenocysteine (Sec60) in SEPHS2 was mutated to cysteine for expression in *E. coli*. VPS37C-2xV5 was subcloned into the NotI and MluI sites of pHAGE-IRES-puro (a gift from Lingling Chen, ShanghaiTech University) for mammalian expression, and the codon-optimized VPS37C, coVPS37C, for *E. coli* was inserted into the EcoRI and SalI sites of pE28a-N-SBPII for *E. coli* expression. TOM1, TOM1L2, TRAPPC3, and VPS29 were subcloned into the EcoRI site of pET28a.

FKBP and FRB were synthesized by Tsingke Biotechnology (Wuhan). FKBP, tagged with a 3×HA, was inserted into the NotI and EcoRI sites of pTol2-EGFPpA, generating pTol2-3×HA-FKBP-EGFPpA. FRB, tagged a 3xV5, and mKate2 were assembled into the NotI and XmaI sites of pTol2-Myc-Halo-PafA-2ApA-CK2, resulting in pTol2-3xV5-FRB-mKate2-PafA-pA. To make a stable cell line, CMV-HA-Halo$^{8KR}$-PafA$^{S126A,K172R}$, SBP$^{K4R}$-sPup$^{K61R}$, TRE3G (a Tet-On promoter), Tet-On 3G transactivator, IRES-EGFP, and Puro were PCR amplified and assembled into pPB-mU6pro (a gift from Zhou Yan, Wuhan University), generating PB-puro-CMV-POST-IT-Tet-SBP$^{K4R}$-sPup$^{K61R}$-iresGFP. The expression of SBP$^{K4R}$-sPup$^{K61R}$ was induced by the application of doxycycline via the Tet-On promoter. The P2A, a 2A ribosome-skipping sequence, was placed between HA-Halo$^{8KR}$-PafA$^{S126A,K172R}$, the Tet-On 3G transactivator, and Puro$^{R}$ to allow for simultaneous expression of the three genes. For the expression of POST-IT in zebrafish, QFGal4, Halo$^{8KR}$-PafA$^{S126A,K172R}$, 5×*QUAS*, EGFP, and SBP$^{K4R}$-sPup$^{K61R}$, were PCR amplified and assembled into the ClaI site of pUbi-QF-CK or pHsp-QF-CK, plasmids developed in our lab for use with the Tol2 transposon system. This assembly resulted in pUbi-QFGal4-2A-POST-IT and pHsp-QFGal4-2A-POST-IT, respectively. The P2A sequence was inserted between QFGal4 and Halo$^{8KR}$-PafA$^{S126A,K172R}$, as well as between EGFP and SBP$^{K4R}$-sPup$^{K61R}$.

## Recombinant protein expression and purification

*E. coli* BL21(DE3) cells were transformed with a plasmid for expression of His-tagged or GST-tagged protein, and cultured in LB medium containing 100 µg/ml of either ampicillin or kanamycin overnight at 37°C until the OD600 reached 0.6–0.7. Protein expression was induced by adding 0.1–0.5 mM IPTG and incubating at 16°C for 16–20 hr. The bacterial culture was then centrifuged at 6000 × *g* for 15 min at 4°C. The pellet was resuspended in *E. coli*-lysis buffer (50 mM Tris-HCl pH 8.0, 300 mM NaCl, 10 mM imidazole, 5% glycerol, 1 mM PMSF), lysed by either sonication or a high-pressure homogenizer (AH-1500, ATS Engineering, China), and subsequently centrifuged at 24,000 × *g* at 4°C for 30 min. PMSF stock solution (200 mM in isopropanol) was freshly added to the appropriate buffer to achieve the final concentration.

For Ni column purification via a 6×His tag, the cleared supernatant was loaded onto Ni Sepharose 6 Fast Flow using a gravity flow. After extensive washing with more than 20 column volume (CV) of Ni-wash buffer (50 mM Tris-HCl pH 8.0, 500 mM NaCl, 10% glycerol, 0.1% Triton X-100, 10 mM imidazole, 1 mM PMSF), the recombinant protein was eluted with 2 CV of elution buffer 1–4 (50 mM Tris-HCl pH 8.0, 200 mM NaCl, varying concentrations of imidazole [250, 300, 350, 400 mM for elution buffer 1–4], 1 mM PMSF). Each fraction's purity was assessed by SDS-PAGE, followed by staining with R-250 Coomassie Brilliant Blue (CBB). The cleaner fractions were collected, concentrated, and stored in storage buffer (50 mM Tris-HCl pH 8.0, 150 mM NaCl, 1 mM DTT, 10% glycerol) through buffer exchange using Amicon Ultra Centrifugal Filter. All protein purification was performed via Ni column purification except for the following cases.

For GST purification via a GST-tag, the cleared supernatant was loaded onto glutathione resin (SA008010, Smart-Lifesciences, China) and allowed to bind at 4°C for 1 hr. The resin was washed with more than 20 CV of Ni-wash buffer, with 10 CV of PreScission Protease reaction buffer (50 mM Tris-HCl pH 8.0, 150 mM NaCl, 2 mM EDTA, 1 mM DTT). 50 U of PreScission Protease (SLP00501, Smart-Lifesciences) was added and incubated at 4°C overnight. The cleaved protein was collected, concentrated, and stored in storage buffer through buffer exchange using Amicon Ultra Centrifugal Filter.

For coVPS37C purification, a protocol using a mutant streptavidin, SAVSBPM18, was applied with minor modifications as previously shown (*Wu et al., 2019*). The synthesized SAVSBPM18 and DBD (dextran binding domain) were subcloned into the NcoI and XhoI sites of pET28a, creating pET-28a-SAVSBPM18-XTEN-DBD, which was then transformed into *E. coli* BL21(DE3) cells. The coVPS37C plasmid was transformed into the *E. coli* ArcticExpress (DE3) strain. The *E. coli* cells were cultured at 37°C until the OD600 reached 0.6, at which point protein expression was induced by adding 0.2 mM IPTG. The *E. coli* cells were subsequently cultured at 12°C overnight and processed to get cleared cell lysates as described above. *E. coli* cell lysates expressing SAVSBPM18-XTEN-DBD were prepared and mixed with coVPS37C lysates at a 1:4 ratio. After 1 hr of incubation at 4°C, the mixture was loaded onto Ni Sepharose 6 Fast Flow resin (17531801, Cytiva, USA), washed with Ni-wash buffer, and eluted with elution buffer 2. The eluate was then mixed with Sephadex-G100 (BS210, Biosharp, China) for 1 hr at 4°C, washed with Ni-wash buffer, and eluted with SBP elution buffer (10 mM biotin, 50 mM Tris-HCl pH 8.0, 300 mM NaCl). The final eluate was collected, concentrated, and stored in storage buffer through buffer exchange using an Amicon Ultra Centrifugal Filter (UFC9050, UFC9030, UFC9010, Millipore, USA). All purified proteins were stored at –80°C until use.

## Cell culture and transfection

HEK293T (CRL-3216, ATCC, USA) and A375 (CRL-1619, ATCC) cells were confirmed to be Mycoplasma-negative and cultured in Dulbecco's modified Eagle's medium (DMEM) (SH30243.01, Cytiva), containing 4.5 g/l glucose, 4 mM glutamine, and 1 mM sodium pyruvate, supplemented with 10% FBS (SA211.02, Cellmax, China), 100 U/ml penicillin, and 100 μg/ml streptomycin at 37°C with 5% $CO_2$. Transient transfections were performed using LipoJet (SL100468, SignaGen, USA), following the manufacturer's instruction. The identity of these cells was authenticated through SRT profiling.

## Fluorescence polarization assay

The FP assay was conducted using a SpectraMax i3x microplate reader (Molecular Devices, USA) equipped with an FP cartridge (Ex. 485 nm, Em. 535 nm) and a grating (G) factor set to 1.5. Black opaque 96-well micro plates (Beyotime, China) were used for all experiments. For the in vitro time course measurement, 30 nM of Halo$^{8KR}$-PafA protein and 2 nM of HaloTag Alexa Fluor 488, Halo-AF488, were prepared in PBS buffer, and the parallel (Iv) and perpendicular (Ih) emission intensities were measured in kinetic mode at 20 s intervals for 30 min. The millipolarization units (mP) were calculated using the formula mP = 1000 × [(Iv – G×Ih) / (Iv + G × Ih)]. In the competition assay with DH derivates, 15 nM of Myc-Halo-PafA and 1 nM of Halo-AF488 were incubated with various DH derivatives in serial dilution at 37°C for 30 min before FP measurement, with Iv and Ih measured at the endpoint. For the cellular binding assay, HEK293T cells were co-transfected with 4 μg of pTol2-myc-Halo-PafA-2ApA-CK2 per well in a 6-well plate and incubated for 48 hr. Different DH derivatives at 1 μM or DMSO were added to each well and incubated for 3 hr. Cell lysates were prepared in PBST (PBS with 0.5% Triton X-100), supplemented with a protease inhibitor cocktail (MB2678, MeilunBio, China), through brief sonication followed by centrifugation. Protein concentration was determined by the BCA assay. Cleared cell lysates (5 μg/ul) were incubated with 2 nM Halo-AF488 in a 60 μl volume at 37°C for 30 min. Iv and Ih were measured, and mP was calculated.

## Western blot analysis

Cells transfected and/or treated with different compounds were harvested at the end of treatment. Cells were washed with ice-cold PBS and lysed with cell lysis buffer that included RIPA buffer (50 mM Tris-HCl pH 7.4, 150 mM NaCl, 1% Triton X-100, 1 mM EDTA, 5% glycerol), supplemented with 20 mM NaF, 2 mM $Na_3VO_4$, 0.4% SDS, 0.2% sodium deoxycholate (DOC), and a protease inhibitor cocktail. Following brief sonication, cell lysates were centrifuged at maximum speed for 10 min at 4°C. The supernatant was collected, and the protein concentration was determined using the BCA assay.

Samples were diluted to the same concentration, mixed with SDS sample buffer for SDS-PAGE, and boiled for 5–10 min. Proteins were separated by 6% or 8% SDS-PAGE and transferred to polyvinylidene fluoride membrane. The membrane was blocked with 5% skim milk in Tris-buffered saline containing 0.05% Tween-20 (TBST) and incubated overnight at 4°C with the specific antibodies diluted in TBST containing 2% BSA: 1:5,000 anti-V5, 1:2500 anti-HA, 1:5000 anti-Myc, 1:5000 anti-SRC, 1:10,000 anti-β-actin, 1:5000 Streptavidin-HRP. After 3–4 TBST washes, the membrane was incubated with a 1:10,000 HRP-conjugated secondary antibody against rabbit or mouse in TBST containing 5% skim milk for 1 hr at room temperature. Following additional washes, the membranes were developed using chemiluminescence with Clarity Western ECL (1705060, Bio-Rad).

## In vitro pupylation assay

For the self-pupylation assay, the reaction consisted of 1 µM Halo-PafA (or a mutant protein) and 10 µM of a Pup substrate in 20 µl of pupylation buffer (PBS supplemented with 15 mM MgCl$_2$ and 10 mM ATP), incubated at 37°C for durations ranging from 10 to 180 min as specified in each figure, and stopped by the addition of 2×SDS sample buffer. Proteins were separated in three layers (4, 6, and 10%) Tricine-SDS-PAGE and stained with CBB. To test the effect of different DH derivatives on the pupylation of the target, 1 µM of a Halo-PafA derivative, 0.5 µM SRC(247-536)-2xV5, and 500 nM of a DH derivative were mixed in the pupylation buffer and incubated at room temperature for 30 min. 10 µM of a Pup substrate was then added and the mixture was incubated at 37°C for another 30 min. The reaction was stopped by adding 2×SDS sample buffer, and the western blot assay was performed using an anti-V5 antibody. In the competition assay, a serial dilution of dasatinib from 0.2 µM to 10 µM was added along with 500 nM DH1 to the reaction before the addition of the Pup substrate. For the identification of pupylated lysine residues in PafA, 2.5 µM PafA and 20 µM HA-Pup were mixed in 50 µl of the pupylation buffer at 37°C for 2 hr. The reaction was stopped by adding 2×SDS sample buffer and separated by SDS-PAGE. CBB-stained bands were excised and analyzed by LC-MS/MS.

## In cellulo pupylation assay

For the western blot assay, HEK293T cells were plated on 6-well plates and co-transfected as described below. For rapamycin-induced pupylation, HEK293T cells were co-transfected with 1 µg of pTol2−3×HA-FKBP-EGFP-pA, 1 µg of pTol2-3xV5-FRB-mKate2-PafA-2ApA-CK2, and 2 µg of pEF6a-HB-Pup (or one of pEF6a-SBP-sPup derivatives). After 24 hr, 100 nM rapamycin was added to induce binding between HA-FKBP-EGFP and V5-FRB-mKate2-PafA. After 18 hr, cells were harvested and lysed with cell lysis buffer. Samples were then analyzed by western blot assay using antibodies against V5 and HA. To evaluate the levels of target labeling by different Halo-PafA variants or the effect of different drug-HTL derivatives, plates or dishes were coated with 10 µg/ml poly-D-lysine, and HEK293T cells were co-transfected with 1 µg of one of Halo-PafA variants, 2.5 µg of pEF6a-SBP$^{K4R}$-sPup$^{K61R}$, and 0.5 µg of pTol2-SRC(247-536)-2xV5. Twenty-four hours later, 250 (or 500 nM) of a DH derivative or 200 nM DC661-H1, unless stated otherwise, was added for an additional 24 hr. Samples were then prepared for western blot analysis. For the competition assay with dasatinib, 100 nM DH1 and a serially increasing concentration of dasatinib, ranging from 0.025 to 25.6 µM with a fourfold increment, were cotreated for 24 hr. For the pulldown assay, cells were plated on 60 mm dishes, co-transfected with 2 µg of one of Halo-PafA variants and 5 µg of pEF6a-SBP$^{K4R}$-sPup$^{K61R}$, and lysed using RIPA buffer (50 mM Tris-HCl pH 7.4, 150 mM NaCl, 1% Triton X-100, 1 mM EDTA, 5% glycerol) supplemented with a protease inhibitor cocktail. To exogenously expression a target protein, 1 µg of a target plasmid, such as SRC(247-536)-2xV5 and VPS37C-2xV5, was included in the transfection. Cleared cell lysates were incubated with 50 µl of streptavidin magnetic beads (L-1012, BioLinkedin) for 2 hr at 4°C. The beads were washed five times with wash buffer A (50 mM Tris-HCl pH 7.4, 400 mM NaCl, 1% Triton X-100, 1 mM EDTA, 5% glycerol). Proteins were eluted with 2×SDS sample buffer and processed for western blot analysis.

## Preparation of proteomics samples for target-ID of DH5

HEK293T cells were plated on two 100 mm dishes for each condition and co-transfected next day with 8 µg of pTol2-HA-Halo$^{8KR}$-PafA$^{S126A,K172R}$ and 16 µg of pEF6a-SBP$^{K4R}$-sPup$^{K61R}$ per dish. After 24 hr, treatments with DMSO, DH5 (250 nM), and a competition containing 250 nM DH5 and 2.5 µM dasatinib were applied and incubated for an additional 24 hr. Cells were harvested in 1 ml of RIPA buffer,

supplemented with a protease inhibitor cocktail for each dish. Following brief sonication, cell debris was removed by centrifugation at maximum speed for 10 min at 4°C. The supernatants were collected and mixed with 200 µl of streptavidin magnetic beads that had been prewashed with RIPA buffer. The beads were washed five times with wash buffer A, then three times with wash buffer B (wash buffer A supplemented with 1 M urea), and eluted twice with 100 µl of 5 mM biotin and 2.5 µM dasatinib in wash buffer B for 1 hr. The eluates were combined and precipitated using trichloroacetic acid (TCA)/DOC. The resultant protein pellets were neutralized with 1 M Tris-HCl pH 8.8, and separated by Tricine-SDS-PAGE, and the CBB-stained bands were excised, with those smaller than 10 kD being removed, and processed for mass analysis.

## Preparation of proteomics samples for target-ID of DC661-H1 using SILAC

A stable HEK293T cells expressing the POST-IT system was generated using the PiggyBac transposon system. Briefly, HEK293T cells on a 60 mm dish were co-transfected with 8 µg of PB-puro-CMV-POST-IT-Tet-SBP$^{K4R}$-sPup$^{K61R}$-iresGFP and 2 µg of the Super PiggyBac Transposase expression vector (PB200PA-1, System Biosciences, USA). Four days later, cells were treated with 5 µg/ml puromycin for about 2 weeks. Media containing puromycin were changed every 3–4 days.

The stable cell line was cultured in SILAC DMEM (280001200, Silantes, Germany) without arginine, lysine, and glutamine, supplemented with 10% dialyzed FBS (C3820-0100, VivaCell Biosciences, China) and 2 mM glutamine. The SILAC medium included 600 mg/l proline (P120032, aladdin, China) to prevent the conversion of arginine to proline (*Bendall et al., 2008*). The heavy medium, K8R10, contained 152.1 mg/l heavy labeled lysine K8 ($^{13}C_6{}^{15}N_2$) (211603902, Silantes) and 44.1 mg/l heavy-labeled arginine R10 ($^{13}C_6{}^{15}N_4$) (201603902, Silantes). The light medium, K0R0, contained 146 mg/l normal lysine K0 (L113006, aladdin) and 42 mg/l normal arginine R0 (1119-34, aladdin). For SILAC experiments, cells were subcultured for at least five-cell passages in either heavy or light DMEM medium.

The expression of POST-IT was induced by the addition of 100 ng/ml doxycycline. Two days later, 500 nM DC661-H1 was added to the heavy medium, or 500 nM DC661-H1 and 5 µM DC660 to the light medium. After 24 hr of incubation, cells were rinsed with ice-cold PBS and collected with 1 ml RIPA buffer, supplemented with a protease inhibitor cocktail, per dish. Cell debris was removed by centrifugation, and protein concentration was determined by the BCA assay. The protein concentration was adjusted to 2 mg/ml, and 200 µl of prewashed streptavidin magnetic beads was added. After about 3 hr of incubation at 4°C, the beads were washed five times with wash buffer A, three times with wash buffer B, and eluted twice with 100 µl of 5 mM biotin and 10 µM DC660 in wash buffer B for 1 hr. The eluates were combined and precipitated using TCA/DOC. The resultant protein pellets were neutralized with 1 M Tris-HCl pH 8.8, and separated by Tricine-SDS-PAGE, and the CBB-stained bands were excised, with those smaller than 10 kD being removed, and processed for mass analysis.

## LC-MS/MS analysis

Excised gels were sliced into small pieces of ~1 mm$^3$, rinsed three times with Milli-Q water, decolorized with 50% acetonitrile and 100 mM $NH_4HCO_3$, and dehydrated with 100% acetonitrile. The dehydrated gel pieces were then reduced with 10 mM DTT in 50 mM $NH_4HCO_3$ at 56°C for 1 hr, carboxymethylated with 55 mM iodoacetamide in the dark at room temperature for 30 min, and digested with trypsin at a 1:50 enzyme to protein mass ratio at 37°C overnight. The peptides were collected, desalted with ZipTipC18 (ZTC18S096, Millipore, USA), dried under vacuum, and stored at –20°C until MS analysis. The resulting tryptic peptides were analyzed by LC-MS/MS at the Institute of Hydrobiology (Chinese Academy of Sciences, Wuhan, China). LC-MS/MS data acquisition was performed using a Q Exactive HF-X mass spectrometer coupled with an Easy-nLC 1200 system (Thermo Fisher Scientific, USA). The peptide mixtures were initially loaded onto a C18 trap column and subsequently separated on a C18 reverse-phase analytical HPLC column, Acclaim PepMap C18 column (75 µm ID ×250 mm, 2 µm particle size, 100 Å pore size, Thermo Fisher Scientific), employing a 100 min gradient program with mobile phase A (0.1% formic acid) and mobile phase B (80% acetonitrile, 0.1% formic acid). The gradient program was as follows: 0–65 min, 5–23% B; 65–85 min, 23–45% B; 85–86 min, 45–90% B; 87–90 min, 90% B; 90–90.1 min, 90–5% B; 90.1–100 min, 5% B, maintained at a constant flow rate of 300 nl/min. For data-dependent acquisition mode analysis,

each scan cycle consisted of one full-scan mass spectrum ($R = 60$ K, AGC = 3e6, max IT = 20 ms, scan range = 350–1800 m/z) followed by 20 MS/MS events ($R = 15$ K, AGC = 2e5, max IT = 50 ms). The higher-energy collisional dissociation collision energy was set to 28 for ion fragmentation. The isolation window for precursor selection was set to 1.6 Da. The former target ion exclusion was set for 25 s.

## Mass spectrometry and data analysis

For the identification of DH5 targets, raw MS data were analyzed using MaxQuant (version 2.1.0.0, Max Planck Institute of Biochemistry; *Cox and Mann, 2008*) with the Andromeda search engine. The MS data were aligned to the UniProt human protein database (Proteome ID: UP000005640). Additional sequences, including Halo[8KR]-PafA[S126A,K172R] and SBP[K4R]-sPup[K61R], were integrated to identify potential proteomics contaminants. Variable modifications included oxidation (M), acetyl (Protein N-term), deamidation (NQ), and GGE (K). The false discovery rate for both peptide and protein identification was set at 1%. The 'Match Between run' option were enabled, with other settings at their default values. Missing values from identified proteins were filled through random imputation and further analyzed statistically to derive p values through unpaired SAM analysis using PANDA-view (*Chang et al., 2018*). Proteins commonly present under DMSO condition were considered nonspecific background interactions and excluded from further analysis.

For the SILAC experiments to identify DC661 targets, raw MS data were analyzed with FragPipe software v20.0 employing the MSFragger search engine (*Kong et al., 2017*). Alignment was performed against the UniProt human protein database (Proteome ID: UP000005640). Halo[8KR]-PafA[S126A,K172R] and SBP[K4R]-sPup[K61R] sequences were incorporated as proteomics contaminants. The SILAC3 workflow was utilized with the default settings for light and heavy SILAC labels. Proteins detected fewer than two times across three biological replicates or exhibiting a high frequency of occurrence (>0.15) in CRAPOME (*Mellacheruvu et al., 2013*) were filtered out from the analysis.

## MST assay

SEPHS2-2xV5, VPS37C-2xV5, VPS29, TRAPPC3, TOM1, and TOM1L2 were labeled using Cy5-NHS ester (A100932, Sangon Biotech, China). Briefly, each target protein was diluted to 10 µM in PBS and underwent buffer exchange with PBS using a Zeba spin desalting column (89882, Thermo Scientific, USA) to remove Tris. The labeling reaction consisted of 80 µl of 10 µM target protein, 10 µl of 1 M NaHCO$_3$ pH 8.3, and 10 µl of 1 mM Cy5-NHS ester in DMSO, and was incubated at 4°C overnight. The next day, unlabeled dyes were removed using Zeba spin desalting columns, and labeling efficiency was assessed by measuring the concentration of Cy5 with a UV-Vis spectrophotometer (SMA5000, Merinton, China). Dasatinib, DH5, HCQ, or DC660 was serially diluted in ligand buffer (PBS with 0.1% Tween-20). Cy5-labeled SEPHS2-2xV5 (5 nM), VPS37C-2xV5 (5 nM), VPS29 (10 nM), TRAPPC3 (10 nM), TOM1 (10 nM), or TOM1L2 (10 nM) was mixed with the corresponding diluted compound in a total volume of 20 µl, and then loaded into Monolith premium capillaries (MO-K025, NanoTemper, Germany). Binding affinity analysis was performed using a NanoTemper Monolith NT.115 instrument (NanoTemper). K$_d$ values were obtained using MO.Affinity Analysis 2.3.0 software (NanoTemper).

## Molecular docking simulation for protein–ligand interaction

The docking simulation was performed using CB-Dock2 (https://cadd.labshare.cn/cb-dock2/index.php), a web server that provides protein–ligand blind docking utilizing Autodock Vina (version 1.1.2). The blind docking simulation was conducted via protein-surface curvature-based cavity detection approach following the website procedures. The SEPHS2 structure information, ma-y60vo.cif, was obtained from the ModelArchive database (https://doi.org/10.5452/ma-y6ovo; *Nunziata et al., 2019*). The PDB files of VPS37 family proteins (Q8NEZ2, VPS37A; Q9H9H4, VPS37B; A5D8V6, VPS37C; Q86XT2, VPS37D) were downloaded from AlphaFold database (https://alphafold.ebi.ac.uk/), and the structure of the yeast ESCRT-I heterotetramer core (2P22) was retrieved from the RCSB protein data bank (https://www.rcsb.org/). Ligand files were downloaded from PubChem as sdf files. The superposition of VPS37C with other VPS37A/B/D proteins was performed using the US-align (Universal Structural alignment) website (https://zhanggroup.org/US-align/; *Zhang et al., 2022*). UCSF ChimeraX 1.6.1 was employed for analyzing, editing, and visualizing the resulting PDB files.

## Cellular thermal shift assay (CETSA)

HEK273T cells were cultured on a 100 mm dish and transfected with 10 μg of VPS37C-2xV5 plasmid. Forty-eight hours later, the cells were washed twice with ice-cold PBS, harvested in 1 ml of PBS buffer supplemented with 0.4% NP-40 and 0.5×protease inhibitor cocktail, incubated for 15 min with gently rotation at room temperature, and centrifuged at maximum speed for 15 min at 4°C. The supernatant was transferred to a new 1.5 ml microtube, and protein concentration was determined by a BCA assay. Cell lysates were diluted to a concentration of 2 mg/ml and divided into two groups. The drug group was treated with 20 μM of DC660 or HCQ, while the control group received the same amount of DMSO. After a 20 min incubation at room temperature, cell lysates were gently mixed and aliquoted into PCR tubes with 40 μl each, then subjected to heat treatment for 4 min, ranging from 47°C to 59°C in 2°C increments, and cooled to room temperature on a T100 thermal cycler (Bio-Rad, USA). Subsequently, cell lysates were transferred to 1.5 ml microtubes and centrifuged at maximum speed for 20 min at 4°C to precipitate unstable insoluble proteins. The supernatant was transferred to a new 1.5 ml microtube, and 10 μl of 5×SDS sample buffer was added to each tube. Samples were boiled at 95°C for 5 min and processed for western blot analysis.

## SRC kinase activity assay

Purified recombinant short SRC(247-536) (0.8 μM), 10 μM ATP, and 40 μM of RR-SRC peptide, RRLIE-DAEYAARG (T510264-0001, Sangon Biotech) were prepared in SRC Kinase Buffer containing 40 mM Tris-HCl pH 7.5, 20 mM $MgCl_2$, 0.1 mg/ml BSA, 2 mM $MnCl_2$, and 50 μM DTT. After a 1 hr incubation at room temperature, ATP levels were determined using the Kinase-Lumi luminescent kinase assay (S0150S, Beyotime) according to the manufacturer's instruction. To measure the $IC_{50}$ values for dasatinib and DH5, dasatinib or DH5 in serial dilution was added to the SRC kinase reaction, and ATP levels were measured as outlined above. Relative SRC activities were calculated as compared to the DMSO condition, set as 100%.

## Zebrafish husbandry

Wild-type AB zebrafish were maintained under standard conditions at 28.5°C, with a 14 hr light and 10 hr dark cycle. Embryos were obtained from several matings, typically involving 1 male and 1–2 females, and were incubated in zebrafish E3 medium (5 mM NaCl, 0.17 mM KCl, 0.33 mM $CaCl_2$, and 0.33 mM $MgSO_4$) at 28.5°C with the same light–dark cycle. The Animal Care and Use Committee of Wuhan University (No. AF078) provided approval for all procedures involving animals.

## In vivo pupylation assay using zebrafish embryos

To express POST-IT in early zebrafish embryos, 20 pg of pUbi-QFGal4-2A-3×Flag-POST-IT plasmid and 25 pg of transposase mRNA were injected into an embryo at the one-cell stage. The next day, GFP-positive and live embryos were selected, dechorionated with Pronase E (HY-114158, MedChemExpress, USA), and divided into two groups of approximately 1000 embryos each. One group was treated with DMSO and the other with 10 μM DH5 for 24 hr. The embryos were then collected, rinsed with PBS, and lysed in cell lysis buffer using a motorized plastic pestle. The resulting lysates were cleared by centrifugation, and protein concentrations were measured using the BCA assay. Samples were diluted to 2 mg/ml in 1 ml, mixed with 200 μl of prewashed streptavidin magnetic beads, and incubated at 4°C for 3 hr. The beads were washed five times with wash buffer A. Proteins were eluted with 2×SDS sample buffer and subsequently processed for western blot analysis.

## Statistical analysis

Data plotting and statistical analysis were conducted using GraphPad Prism 9.5 software (GraphPad Software, Inc). Data are presented as mean ± s.e.m. or mean ± s.d., as specified in each figure and its legend. p values were determined using Student's unpaired two-sided *t*-test, unless otherwise mentioned. $p < 0.05$ was considered a statistical difference in this study.

## Chemical synthesis

All reagents and solvents were obtained from commercial sources and used without further purification. The chemical reactions were monitored by thin-layer chromatography (TLC). [1]H and [13]C NMR spectra were recorded on AV-400 spectrometer (Bruker instrument, USA). HRMS measurements were

performed with an Agilent QTOF 6520 mass spectrometer (Agilent Technologies, USA) with electrospray ionization (ESI) as the ion source. The purity of all target compounds was analyzed by HPLC (254 nm wavelength in an Agilent LC-1220 instrument) via a C18 column (5 µm, 4.6 mm × 150 mm).

## Ethyl 6-((2-(4-(6-((5-((2-chloro-6-methylphenyl)carbamoyl)thiazol-2-yl)amino)-2-methylpyrimidin-4-yl)piperazin-1-yl)ethyl)amino)-6-oxohexanoate (3)

A mixture of compound **1** (350 mg, 0.72 mmol), compound **2** (150 mg, 0.86 mmol), HATU (328 mg, 0.86 mmol), and 1 ml of DIPEA in DMF (10 ml) was stirred at room temperature for 2 hr. The mixture was diluted with $H_2O$ (20 ml) and extracted with EtOAc (20 ml ×3). The organic layer was dried and concentrated. The residue was purified by column chromatography (DCM:MeOH = 20:1 to 15:1) to yield a white solid 210 mg. Yield: 45%. $^1H$ NMR (400 MHz, DMSO-$d_6$) δ 11.59 (s, 1H), 10.08 (s, 1H), 8.33 (s, 1H), 7.91 (s, 1H), 7.39 (d, $J$ = 7.6 Hz, 1H), 7.27–7.21 (m, 1H), 6.13 (s, 1H), 4.03 (q, $J$ = 7.1 Hz, 2H), 3.55–3.44 (m, 4H), 3.27–3.15 (m, 2H), 2.49–2.33 (m, 9H), 2.31–2.22 (m, 5H), 2.11–2.03 (m, 2H), 1.53–1.44 (m, 4H), 1.16 (t, $J$ = 7.1 Hz, 3H) (*Figure 2—figure supplement 1*).

## N1-(2-(4-(6-((5-((2-Chloro-6-methylphenyl)carbamoyl)thiazol-2-yl)amino)-2-methylpyrimidin-4-yl)piperazin-1-yl)ethyl)-N6-(18-chloro-3,6,9,12-tetraoxaoctadecyl)adipamide (DH1)

To a mixture of compound **3** (141 mg, 0.22 mmol) in THF: $H_2O$ (2:1, 5 ml), NaOH was added (20 mg). The mixture was stirred at room temperature for 2 hr, then concentrated under vacuum to afford compound **4**, which was used in the next step without further purification. A mixture of compound **4** (0.22 mmol), compound **7** (0.53 mmol), HATU (100 mg, 0.27 mmol), and 0.2 ml DIPEA in DCM:DMF (2:1, 9 ml) was stirred at room temperature for 2 hr. The mixture was diluted with $H_2O$ (20 ml) and extracted with DCM (20 ml ×3). The organic layers were dried and concentrated. The residue was purified by column chromatography (DCM:MeOH = 10:1 to 5:1) to yield a white solid 23 mg. Yield: 11% (*Figure 2—figure supplement 1*). $^1H$ NMR (400 MHz, Methanol-$d_4$) δ 8.58 (d, $J$ = 4.4 Hz, 1H), 8.27 (d, $J$ = 8.0 Hz, 1H), 8.18 (s, 1H), 7.39 (dd, $J$ = 13.8, 5.5 Hz, 2H), 7.26 (q, $J$ = 8.0 Hz, 2H), 6.06 (s, 1H), 3.77–3.71 (m, 4H), 3.70–3.44 (m, 18H), 3.37 (t, $J$ = 5.3 Hz, 2H), 2.80 (s, 4H), 2.84–2.76 (m, 2H), 2.76–2.70 (s, 3H), 2.41–2.18 (m, 9H), 1.81–1.72 (m, 2H), 1.69–1.56 (m, 6H), 1.52–1.39 (m, 4H), 1.24–1.12 (m, 2H). HRMS (ESI) calcd for $[C_{42}H_{63}Cl_2N_9O_7SNa]^+$ $[M+Na]^+$, 930.3846; found 930.3843 (*Figure 2—figure supplement 2*).

## 2-(2-((6-Chlorohexyl)oxy)ethoxy)ethan-1-amine (8)

A mixture of compound **6** (2 g, 19.02 mmol) in DMF (10 ml) was added to NaH (1.14 g, 28.53 mmol) and stirred at 0°C for 30 min. Then, compound **7** (5.6 g, 22.82 mmol) in 10 ml DMF was added slowly. The mixture was stirred at 0°C until the reaction was complete, as monitored by TLC. The reaction was quenched by adding 40 ml of saturated $NH_4Cl$ solution and then extracted with EtOAc (30 ml ×3). The organic layers were dried and concentrated. The residue was purified by column chromatography (DCM:MeOH = 20:1 to 15:1) to yield a colorless oil. Yield: 67%. $^1H$ NMR (400 MHz, CDCl$_3$) δ 3.65–3.56 (m, 4H), 3.56–3.50 (m, 4H), 3.47 (t, $J$ = 6.7 Hz, 2H), 2.89 (t, $J$ = 4.7 Hz, 2H), 2.59 (s, 3H), 1.85–1.72 (m, 2H), 1.66–1.54 (m, 2H), 1.53–1.30 (m, 4H) (*Figure 4—figure supplement 2*).

### Activation of glycol analogs (11a–c)

Glycol analogs **9a–c** (20 mmol) were added to a mixture of compound **10** (24 mmol) in 30 ml of DCM. Then, Et$_3$N (30 mmol) was slowly added to the mixture at 0°C. The mixture was then warmed to room temperature and stirred overnight. The reaction was diluted by adding 40 ml $H_2O$ and extracted with DCM (30 ml ×3). The organic layers were dried and concentrated. The residue was purified by column chromatography (PE:EtOAc = 4:1 to 3:1) to yield a yellowish oil. Yield: 81–92%. $^1H$ NMR of **11c** (400 MHz, CDCl$_3$) δ 8.32–8.23 (m, 4H), 7.43–7.34 (m, 4H), 4.50–4.43 (m, 4H), 3.89–3.82 (m, 4H), 3.77 (s, 4H) (*Figure 4—figure supplement 2*).

## Monosubstituted Halo ligands (12a–c)

Activated glycol analogs **11a–c** (10 mmol) and compound **8** (9 mmol) in 10 ml DMF were stirred at room temperature until reaction was complete, as monitored by TLC. The reaction was diluted by adding 20 ml of saturated aqueous $NH_4Cl$ and extracted with EtOAc (30 ml ×3). The organic layers were dried and concentrated. The residue was purified by column chromatography (PE: EtOAc = 1:1) to yield a yellowish oil. Yield: 55–69%. $^1H$ NMR of **12c** (400 MHz, CDCl$_3$) δ 8.30–8.20 (m, 2H), 7.41–7.33 (m, 2H), 4.46–4.38 (m, 2H), 4.28–4.16 (m, 2H), 3.83–3.76 (m, 2H), 3.72–3.61 (m, 6H), 3.60–3.47 (m, 9H), 3.46–3.39 (m, 2H), 3.34 (t, $J$ = 5.3 Hz, 2H), 1.81–1.68 (m, 2H), 1.64–1.51 (m, 2H), 1.49–1.27 (m, 4H) (*Figure 4—figure supplement 2*).

## Synthesis of dasatinib-Halo derivatives 2–5 (DH2–5)

Dasatinib ligands were prepared as previously reported with some modification (*Wang et al., 2015*). Briefly, dasatinib was activated by reacting with compound **10**, followed by the addition of diamine ligands **15a** and **15b** to form **16a** and **16b**. The reactions were diluted by adding 20 ml of saturated aqueous $NH_4Cl$ and extracted with EtOAc (30 ml ×3). Without further purification, they were reacted with monosubstituted halo ligands **12a–c** to afford **DH2–5** with diverse linker lengths (off-white solid, yields: 20–35% over three steps) (*Figure 4—figure supplement 2*).

## 17-Chloro-4-oxo-3,8,11-trioxa-5-azaheptadecyl(2-(4-(6-((5-((2-chloro-6-methylphenyl)carbamoyl)thiazol-2-yl)amino)-2-methylpyrimidin-4-yl)piperazin-1-yl)ethyl) ethane-1,2-diyldicarbamate (DH2)

$^1H$ NMR (400 MHz, CD$_3$OD) δ 8.19 (s, 1H), 7.36 (dd, $J$ = 7.3, 2.2 Hz, 1H), 7.30–7.19 (m, 2H), 6.01 (s, 1H), 4.24 (t, $J$ = 5.5 Hz, 2H), 4.16 (t, $J$ = 4.7 Hz, 4H), 3.78–3.63 (m, 8H), 3.63–3.51 (m, 8H), 3.47 (t, $J$ = 6.5 Hz, 2H), 3.29 (t, $J$ = 5.5 Hz, 2H), 3.23 (d, $J$ = 6.9 Hz, 4H), 2.72 (t, $J$ = 5.5 Hz, 2H), 2.65 (t, $J$ = 5.1 Hz, 4H), 2.49 (s, 3H), 2.34 (s, 3H), 1.81–1.70 (m, 2H), 1.64–1.53 (m, 2H), 1.51–1.37 (m, 4H). HRMS (ESI) calcd for [C$_{39}$H$_{57}$Cl$_2$N$_{10}$O$_9$S]$^+$ [M+H]$^+$ 911.3408, found 911.3416. HPLC: $t_R$ = 5.59 min, purity = 100% (*Figure 4—figure supplement 3*).

## 17-Chloro-4-oxo-3,8,11-trioxa-5-azaheptadecyl(2-(4-(6-((5-((2-chloro-6-methylphenyl)carbamoyl)thiazol-2-yl)amino)-2-methylpyrimidin-4-yl)piperazin-1-yl)ethyl) ethane-1,2-diyldicarbamate (DH3)

$^1H$ NMR (400 MHz, CD$_3$OD) δ 8.19 (s, 1H), 7.41–7.33 (m, 1H), 7.31–7.21 (m, 2H), 6.06 (s, 1H), 4.33–4.25 (m, 2H), 4.22 (m, 4H), 3.64–3.44 (m, 10H), 3.30 (t, $J$ = 5.5 Hz, 2H), 3.27–3.18 (m, 12H), 2.93–2.85 (m, 2H), 2.82 (m, 4H), 2.50 (s, 3H), 2.35 (s, 3H), 1.82–1.70 (m, 2H), 1.59 (m, 2H), 1.53–1.35 (m, 4H). HRMS (ESI) calcd for [C$_{41}$H$_{60}$Cl$_2$N$_{10}$O$_{10}$SNa]$^+$ [M+Na]$^+$ 977.3489, found 977.3497. HPLC: $t_R$ = 5.68 min, purity = 100% (*Figure 4—figure supplement 4*).

## 23-Chloro-10-oxo-3,6,9,14,17-pentaoxa-11-azatricosyl(2-(4-(6-((5-((2-chloro-6-methylphenyl)carbamoyl)thiazol-2-yl)amino)-2-methylpyrimidin-4-yl)piperazin-1-yl)ethyl) ethane-1,2-diyldicarbamate (DH4)

$^1H$ NMR (400 MHz, CD$_3$OD) δ 8.18 (s, 1H), 7.37 (dd, $J$ = 7.2, 2.2 Hz, 1H), 7.30–7.20 (m, 2H), 6.01 (s, 1H), 4.24 (t, $J$ = 5.5 Hz, 2H), 4.17 (t, $J$ = 4.7 Hz, 4H), 3.71–3.62 (m, 12H), 3.62–3.50 (m, 8H), 3.47 (t, $J$ = 6.6 Hz, 2H), 3.33–3.26 (m, 3H), 3.22 (s, 3H), 2.71 (t, $J$ = 5.6 Hz, 2H), 2.64 (t, $J$ = 4.9 Hz, 4H), 2.49 (s, 3H), 2.34 (s, 3H), 1.82–1.70 (m, 2H), 1.59 (m, 2H), 1.53–1.33 (m, 4H). HRMS (ESI) calcd for [C$_{43}$H$_{64}$Cl$_2$N$_{10}$O$_{11}$SNa]$^+$ [M+Na]$^+$ 1021.3751, found 1021.3751. HPLC: $t_R$ = 5.59 min, purity = 100% (*Figure 4—figure supplement 5*).

## 23-Chloro-10-oxo-3,6,9,14,17-pentaoxa-11-azatricosyl(2-(4-(6-((5-((2-chloro-6-methylphenyl)carbamoyl)thiazol-2-yl)amino)-2-methylpyrimidin-4-yl)piperazin-1-yl)ethyl) pentane-1,5-diyldicarbamate (DH5)

$^1H$ NMR (400 MHz, CD$_3$OD) δ 8.19 (s, 1H), 7.36 (dd, $J$ = 7.3, 2.3 Hz, 1H), 7.30–7.19 (m, 2H), 6.00 (s, 1H), 4.22 (t, $J$ = 5.5 Hz, 2H), 4.16 (dd, $J$ = 6.1, 3.4 Hz, 4H), 3.73–3.61 (m, 12H), 3.61–3.50 (m, 8H), 3.47 (t, $J$ = 6.6 Hz, 2H), 3.29 (t, $J$ = 5.5 Hz, 2H), 3.11 (m, 4H), 2.69 (t, $J$ = 5.5 Hz, 2H), 2.62 (t, $J$ = 5.1 Hz, 4H), 2.48

(s, 3H), 2.34 (s, 3H), 1.82–1.70 (m, 2H), 1.64–1.26 (m, 12H). HRMS (ESI) calcd for $[C_{46}H_{70}Cl_2N_{10}O_{11}SNa]^+$ $[M+Na]^+$ 1063.4221, found 1063.4229. HPLC: $t_R$ = 5.60 min, purity = 100% (**Figure 4—figure supplement 6**).

### N$^1$-(7-Chloroquinolin-4-yl)-N$^6$-(6-((7-chloroquinolin-4-yl)amino)hexyl)hexane-1,6-diamine (21, DC660)

A mixture of compound **19** (25 mmol), **20** (10 mmol), Pd(OAc)$_2$ (0.2 mmol), BINAP (0.4 mmol), and K$_3$PO$_4$ (30 mmol) in 20 ml of 1,4-dioxane was stirred at 100°C overnight under an Ar atmosphere. The reaction was diluted by adding 20 ml of saturated aqueous NH$_4$Cl and extracted with EtOAc (30 ml ×3). The organic layer was dried and concentrated. The residue was purified by column chromatography (DCM:MeOH = 10:1 to 5:1) to yield a white solid. Yield: 83% (**Figure 6—figure supplement 2**). $^1$H NMR (400 MHz, CD$_3$OD) δ 8.36 (d, J = 5.6 Hz, 2H), 8.12 (d, J = 9.0 Hz, 2H), 7.79 (d, J = 2.2 Hz, 2H), 7.41 (dd, J = 9.0, 2.2 Hz, 2H), 6.53 (d, J = 5.7 Hz, 2H), 3.39 (t, J = 7.2 Hz, 4H), 2.76–2.65 (m, 4H), 1.84–1.72 (m, 4H), 1.65–1.40 (m, 12H) (**Figure 6—figure supplement 3**).

### Tert-butyl *bis*(6-((7-chloroquinolin-4-yl)amino)hexyl)glycinate (23)

A mixture of compound **21** (10 mmol), **22** (12 mmol), and K$_2$CO$_3$ (15 mmol) in 20 ml of DMF was stirred at 80°C overnight under an Ar atmosphere. The reaction was diluted by adding 20 ml of saturated aqueous NH$_4$Cl and extracted with EtOAc (30 ml ×3). The organic layer was dried and concentrated. The residue was purified by column chromatography (DCM:MeOH = 50:1 to 20:1) to yield an off-white solid. Yield: 79%. $^1$H NMR (400 MHz, CDCl$_3$) δ 8.40 (d, J = 5.9 Hz, 2H), 8.19 (d, J=9.0 Hz, 2H), 8.00 (d, J = 2.1 Hz, 2H), 7.35 (dd, J=9.0, 2.1 Hz, 2H), 6.83 (s, 2H), 6.44 (d, J = 5.9 Hz, 2H), 3.40 (q, J = 6.7 Hz, 4H), 3.20 (s, 2H), 2.54 (t, J = 7.0 Hz, 4H), 1.76 (m, 4H), 1.52–1.24 (m, 21H) (**Figure 6—figure supplement 2**).

### Synthesis of DC661-Halo derivative 1 (DC661-H1)

The deprotection of compound **23** (5 mmol) was carried out by adding 2 ml of TFA. The TFA was then evaporated under vacuum to afford compound **24**. The residue was used directly for the next step. A mixture of **24**, **15b** (5 mmol), HATU (6 mmol), and DIPEA (20 mmol) in 10 ml of DMF was stirred at room temperature for 1 hr. The reaction was then diluted by adding 20 ml of saturated aqueous NH$_4$Cl and extracted with EtOAc (30 ml ×3). The organic layer was dried and concentrated. The residue was further reacted with **12c** (6 mmol) for 2 hr. The reaction was again diluted by adding 20 ml of saturated aqueous NH$_4$Cl and extracted with EtOAc (30 ml ×3). The organic layer was dried and concentrated. The residue was purified by column chromatography (DCM:MeOH = 30:1 to 20:1) to yield a yellowish solid. Yield: 43% over three steps (**Figure 6—figure supplement 2**). $^1$H NMR (400 MHz, CD$_3$OD) δ 8.37 (d, J = 6.3 Hz, 2H), 8.24 (d, J = 9.0 Hz, 2H), 7.80 (d, J = 2.1 Hz, 2H), 7.52 (dd, J = 9.1, 2.1 Hz, 2H), 6.67 (d, J = 6.3 Hz, 2H), 4.16 (dq, J = 12.8, 4.7 Hz, 6H), 3.73–3.42 (m, 20H), 3.29 (q, J = 5.8 Hz, 4H), 3.21 (t, J = 7.3 Hz, 2H), 3.12 (s, 2H), 3.10–2.99 (m, 2H), 2.56 (t, J = 7.4 Hz, 4H), 1.67–1.23 (m, 28H). $^{13}$C NMR (151 MHz, CD$_3$OD) δ 172.34, 157.45, 157.45, 153.48, 153.48, 147.00, 147.00, 143.81, 143.81, 136.98, 136.98, 125.75, 125.75, 123.68, 123.68, 122.84, 122.84, 116.47, 116.47, 98.24, 98.24, 72.29, 70.79, 70.78, 70.18, 70.15, 70.00, 69.84, 69.75, 69.52, 69.18, 69.16, 63.67, 63.56, 60.81, 57.59, 55.15, 44.31, 44.29, 42.97, 40.28, 40.21, 38.46, 32.32, 29.12, 29.10, 28.83, 27.80, 26.75, 26.63, 26.58, 26.31, 25.05, 23.73, 21.12. HRMS (ESI) calcd for $[C_{55}H_{83}Cl_3N_8O_9Na]^+$ $[M+Na]^+$ 1127.5246, found 1127.5236.

## Materials availability statement

Plasmids and cell lines generated in this study are available upon a reasonable request from the corresponding author (youngnam_jin@whu.edu.cn).

## Acknowledgements

We thank the staff in the core facility of the Medical Research Institute at Wuhan University for their technical support with equipment, and the Institute of Hydrobiology for their support in proteomics. We are also grateful to Dr. Min Zhuang and Dr. Yang Yu for providing the PafA and HaloTag7 plasmids, respectively. This work was supported by the National Natural Science Foundation of China

(32070832 and 32150610476 to YNJ; 82273774 to H-BZ) and the Fundamental Research Funds for the Central Universities (2042022dx0003 to YVY).

## Additional information

### Funding

| Funder | Grant reference number | Author |
| --- | --- | --- |
| National Natural Science Foundation of China | 32070832 | Youngnam N Jin |
| National Natural Science Foundation of China | 32150610476 | Youngnam N Jin |
| National Natural Science Foundation of China | 82273774 | Hai-bing Zhou |
| Fundamental Research Funds for the Central Universities | 2042022dx0003 | Yanxun V Yu |

The funders had no role in study design, data collection and interpretation, or the decision to submit the work for publication.

### Author contributions

Yingjie Sun, Changheng Li, Formal analysis, Investigation, Methodology, Writing – original draft; Xiaofei Deng, Investigation, Methodology, Writing – original draft; Wenjie Li, Weiqi Ge, Miaoyuan Shi, Ying Guo, Investigation; Xiaoyi Deng, Investigation, Visualization; Yanxun V Yu, Supervision, Funding acquisition, Writing – original draft, Project administration, Writing – review and editing; Hai-bing Zhou, Supervision, Funding acquisition, Methodology, Writing – original draft, Project administration, Writing – review and editing; Youngnam N Jin, Conceptualization, Formal analysis, Supervision, Funding acquisition, Investigation, Visualization, Methodology, Writing – original draft, Project administration, Writing – review and editing

### Author ORCIDs

Yanxun V Yu https://orcid.org/0000-0001-6617-0166
Youngnam N Jin https://orcid.org/0009-0007-2632-3237

### Ethics

The Animal Care and Use Committee of Wuhan University (No. AF078) provided approval for all procedures involving animals.

Reviewer #2 (Public review): https://doi.org/10.7554/eLife.102667.3.sa1
Reviewer #3 (Public review): https://doi.org/10.7554/eLife.102667.3.sa2
Author response https://doi.org/10.7554/eLife.102667.3.sa3

## Additional files

### Supplementary files

MDAR checklist

### Data availability

All data generated or analyzed during this study are included in the manuscript and supporting files. This study includes no data deposited in external repositories.

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
