## [Editor Report · eLife Assessment]

The study presents **important** findings that reveal SEPHS2 and VPS37C as new potential drug targets for dasatinib and hydroxychloroquine respectively in addition to confirming known targets of these drugs. The evidence provided is **compelling** as observed in the methods, data, and analyses. This article will be of great interest to chemical biologists, biochemists, and scientists in drug discovery and diagnostics.

---

## [Referee Report · Reviewer #2 (Public review)]

Summary:

The study by Sun et al. introduces a useful system utilizing the proteasomal accessory factor A (PafA) and HaloTag for investigating drug-protein interactions in both in vitro (cell culture) and in vivo (zebrafish) settings. The authors presented the development and optimization of the system, as well as examples of its application and the identification of potential novel drug targets. However, the manuscript requires considerable improvements, particularly in writing and justification of experimental design. There are several inaccuracies in data description and a lack of statistics in some figures, undermining the conclusions drawn in the manuscript. Additionally, the authors introduced variants of the ligands and its cognate substrates, yet their use in different experiments appears random and lacks justification. It is challenging for readers to remember and track the specific properties of each variant, further complicating the interpretation of the results.

The conclusions of this paper are mostly backed by data, but certain aspects of data analysis and description require further clarification and expansion.

Comments on revisions:

We would like thank authors for submitting this revised version. We appreciate their inclusion of additional experiments, which convincingly demonstrate the absence of significant toxicity for in vivo applications. All our concerns and questions have been fully addressed. The clarity and quality of the writing have been substantially improved. We believe this innovative proximity labeling tool would be inspiring and valuable for the field.

---

## [Referee Report · Reviewer #3 (Public review)]

Summary:

This manuscript introduces POST-IT (Pup-On-target for Small molecule Target Identification Technology), a novel non-diffusive proximity tagging system for identifying target proteins in live cells and organisms. This technology preserves cellular context essential for capturing specific drug-protein interactions, including transient complexes and membrane-associated proteins. Using an engineered fusion of proteasomal accessory factor A (PafA) and HaloTag, POST-IT specifically labels proximal proteins upon binding to a small molecule, with extensive optimization to enhance specificity and efficiency.

Strengths:

The study successfully identifies known targets and discovers new binders, such as SEPHS2 for dasatinib and VPS37C for hydroxychloroquine, advancing our understanding of their mechanisms. Additionally, its application in live zebrafish embryos demonstrates POST-IT's potential for widespread use in biological research and drug development.

Comments on revisions:

The authors have addressed most of the issues I raised in my review. I have no further comments.

---

## [Author Response]

The following is the authors’ response to the original reviews.

**Public Reviews:**

**Reviewer #1 (Public review):**
(1) The technology requires a halo-tagged derivation of the active compound, and the linked position will have a huge impact on the potential "target hits" of the molecules. Given the fact that most of the active molecules lack of structure-activity relationship information, it is very challenging to identify the optimal position of the halo tag linkage.

We appreciate your insightful comment. While finding the optimal position to attach a chemical linker to a small molecule of interest is indeed a challenging but necessary step, this is a common difficulty across all target-ID methods, except for those that are modification-free, as we described in Discussion. However, modification-free approaches such as DARTS, CETSA, and TPP have their own limitations, such as low sensitivity and a high false-positive rate. Additionally, DARTS and SPROX are limited to use with cell lysates. Please refer to the introduction in our manuscript for more details on these approaches. On the other hand, synthesizing HTL derivatives is relatively straightforward compared to other modifications, and we provide helpful guidelines for chemical linker design, provided the optimal chemical moiety has been identified, which is crucial for target identification. We selected dasatinib and HCQ/CQ as model compounds because previous studies offered insights into their derivative synthesis. Our data also show that DH5 retains strong kinase inhibitory activity (Figure 4—figure supplement 2), and DC661-H1 demonstrates potent inhibition of autophagy (Figure 6—figure supplement 1). For novel compounds, conducting a thorough structure-activity relationship (SAR) study is essential to determine the optimal position for HTL derivative synthesis.

(2) Although POST-IT works in zebrafish embryos, there is still a long way to go for the broad application of the technology in other animal models.

Thank you for your constructive comment. Yes, there is still a long way to go in developing the POST-IT system for broader applications in other animal models, especially in mice. However, we hope that our study provides valuable insights and inspiration to scientists and experts for applying the POST-IT system in various models. We are also committed to further improving its applicability.

(3) The authors identified SEPHS2 as a new potential target of dasatinib and further validated the direct binding of dasatinib with this protein. However, considering the super strong activity of dasatinib against c-Src (sub nanomolar IC50 value), it is hard to conclude the contribution of SEPHS2 binding (micromolar potency) to its antitumor activity.

Thank you for your insightful comment. We agree that the anticancer activity of dasatinib primarily results from inhibiting tyrosine kinases such as SRC and ABL. However, SEPHS2 contains an “opal" termination codon, UGA, at the 60th amino acid residue, which codes for selenocysteine. Due to the technical challenge of expressing selenoproteins in *E. coli*, we mutated it to cysteine for expression in *E. coli* to avoid premature translation termination, as described in the Materials and Methods section. Although the purified recombinant SEPHS2 shows a Kd of about 10 µM for dasatinib, the binding affinity to endogenous SEPHS2 may be higher since selenocysteine is larger and more electronegative than cysteine. This presents an interesting area for future investigation. Furthermore, our study of dasatinib’s binding to SEPHS2 could help facilitate the development of new SEPHS2 inhibitors, potentially targeting the active site of SEPHS2.

**Reviewer #3 (Public review):**
(1) Target Specificity: It is crucial for the authors to differentiate between the primary targets of the POST-IT system and those identified as side effects. This distinction is essential for assessing the specificity and utility of the technology.

Thank you for your insightful comment. Drugs inevitably bind to various proteins with differing affinities, which can contribute to both side effects and beneficial outcomes. Typically, the primary targets exhibit high affinities. In this manuscript, we ranked the identified protein targets of DH5 based on affinity from mass spectrometry and p-values (Fig. 5A), and for DC661-H1, we used the SILAC ratio (Fig. 6A). We also individually assessed many drug-protein binding affinities using the MST assay, as well as in vitro and in cellulo assays, demonstrating their specificity. Moreover, we believe it is essential to identify as many protein targets as possible at physiological drug concentrations to better understand the drug’s side effects. Of course, further investigation is required to assess the roles and effects of these target proteins.

(2) In Vivo Target Identification: The manuscript lacks detailed clarity on which specific targets were successfully identified in the in vivo experiments. Expanding on this information would provide a clearer view of the system's effectiveness and scope in complex biological settings.

Thank you for your insightful comment regarding in vivo target identification. In this manuscript, we utilized a cell line as the primary method for in vivo target identification and validation after optimizing our system in test tubes. We successfully validated many of the targets identified using our POST-IT system (Figure 6—figure supplement 3). To demonstrate the proof of principle for in vivo application, we employed zebrafish embryos as an in vivo model, showing that endogenous SRC can be effectively pulled down by DH5 treatment (Fig. 7). While we could have explored the entire proteome to identify endogenous target proteins in zebrafish that bind to DH5 or dasatinib, we felt this would extend beyond our original scope, given that we have already demonstrated POST-IT’s ability to identify target proteins for dasatinib. Specific target identification and validation are crucial when using zebrafish for drug discovery. Additionally, we acknowledge that drugs likely interact with a range of protein targets in living organisms and may undergo metabolism and interactions within the circulatory system, which we address in our discussion.

(3) Reproducibility and Scalability: Discussion on the reproducibility of the POST-IT system across various experimental setups and biological models, as well as its scalability for larger-scale drug discovery programs, would be beneficial.

Thank you for the suggestion. While our system has shown high reproducibility in our experiments, further improving both reproducibility and scalability would be advantageous. One potential approach to address this is through the generation of stable-expressing cell lines and transgenic zebrafish lines, which we have discussed in the revised manuscript. Establishing stable cell lines with robust POST-IT expression could enhance scalability for drug discovery applications.

(4) Quantitative Analysis: A more detailed quantitative analysis of the protein interactions identified by POST-IT, including statistical significance and comparative data against other technologies, would enhance the manuscript.

Thank you for your suggestion. In our assessment of drug-protein affinity, we included Kd values as quantitative measures using MST assays. The protein targets of dasatinib identified through mass spectrometry are also accompanied by p-values for quantitative analysis (Fig. 5A), and the detailed procedures are described in the Material and methods section. While it is challenging to provide direct comparative data against other technologies, our system successfully identified many known target proteins for dasatinib, as well as SEPHS2 and VPS37C as new targets for dasatinib and for HCQ/CQ, respectively, which were not detected by other methods.

(5) Technological Limitations: The authors should discuss any limitations or potential pitfalls of the POST-IT system, which would be crucial for future users and for guiding subsequent improvements.

Thank you for your insightful suggestion We agree that clearly defining the technological limitations is important. Therefore, we have expanded our original discussion on the limitations of our POST-IT system (Discussion section, paragraph 6).

(6) Long-Term Stability and Activity: Information on the long-term stability and activity of the POST-IT components in different biological environments would ensure the reliability of the system in prolonged experiments.

Yes, this is an important question. We did not notice any stability or toxicity issues with Halo-PafA and Pup substrates in HEK293T cells or zebrafish, which is an important factor for stable cell lines and transgenic zebrafish lines. However, HTL derivatives of the drug could be toxic or unstable due to the nature of the drug or its metabolism, which needs to be taken into account when designing experiments, and we have included this in the Discussion.

(7) Comparison with Existing Technologies: A detailed comparison with existing proximity tagging and target identification technologies would help position POST-IT within the current landscape, highlighting its unique advantages and potential drawbacks.

We appreciate your valuable feedback and agree that such comparisons are crucial. We have included a detailed overview and comparison of existing proximity-tagging systems and their related target identification technologies in the Introduction (lines 78-100) and Discussion (lines 391-412), highlighting their respective pros and cons. Additionally, we have expanded the discussion to further compare these technologies with our POST-IT system, addressing its advantages and limitations (lines 378-390, lines 448-467). We hope this provides sufficient context and information to effectively position POST-IT among the landscape of proximity-tagging target identification technologies.

(8) Concerns Regarding Overexposed Bands: Several figures in the manuscript, specifically Figure 3A, 3B, 3C, 3F, 3G, Figure 4D, and the second panels in Figure 7C as well as some figures in the supplementary file, exhibit overexposed bands.

We appreciate your astute observation regarding the overexposed bands and apologize for any confusion. The “overexposed” bands represent the unpupylated proteins, while the bands above them correspond to the pupylated proteins. We intended to clearly show both pupylated and unpupylated bands, although the latter are generally much weaker. We are currently working on further improving our POST-IT system to enhance pupylation efficiency.

(9) Innovation Concern: There is a previous paper describing a similar approach: Liu Q, Zheng J, Sun W, Huo Y, Zhang L, Hao P, Wang H, Zhuang M. A proximity-tagging system to identify membrane protein-protein interactions. Nat Methods. 2018 Sep;15(9):715-722. doi: 10.1038/s41592-018-0100-5. Epub 2018 Aug 13. PMID: 30104635. It is crucial to explicitly address the novel aspects of POST-IT in contrast to this earlier work.

Thank you for bringing this to our attention. Proximity-tagging systems like BioID, TurboID, NEDDylator, and PafA (Lui Q et al., Nat Methods 2018) were initially developed to study protein-protein interactions or identify protein interactomes, as these applications are of broader interest and generally easier to implement. However, applying proximity-tagging systems for small molecule target identification requires significant optimization. As described in the introduction (lines 78-100), target protein identification systems have since been developed using TurboID and NEDDylator (Tao AJ et al., Nat Commun 2023; Hill ZB et al., J Am Chem Soc 2016). It is conceivable that a PafA-based proximity-tagging system could also be adapted for target-ID, and other groups may pursue this approach in the future. Although the PafA-Pup system shows great promise for target-ID applications, extensive optimization was needed to enable its use for this purpose. Finally, we demonstrate that POST-IT offers distinct advantages over other proximity-tagging-based target-ID systems. For more details, please refer to the introduction and discussion sections.

**Recommendations for the authors:**

**Reviewer #2 (Recommendations for the authors):**
(1) Figure 1- Figure Supplement 1A: The Pup substrate "HB-Pup" is mentioned, but the main text or figure legend provides no introduction or description.

We appreciate your astute observation. We have added a description in the main text and figure legend as follows: “…and used HB-Pup as a control, which contains 6´His and BCCP at the N terminus of Pup” in the main text (line 142) and HB, TS, and SBP refer to 6´His and BCCP, twin-STII (Strep-tag II), and streptavidin binding peptide, respectively.” in the Figure 1-figure supplement 1A.

(2) Figure 1 - Figure Supplement 3B: The authors used TS-sPupK61R as a substrate but did not explain why. The main text mentions that mutating sPup alone did not affect polypupylation, raising the question of why TS-sPupK61R was used in this figure. Furthermore, while the authors state that polypupylation becomes evident after 1 hour of incubation (more pronounced after 2 or 3 hours), the reactions here were conducted for only 30 minutes.

Thank you for your question. Figure 1 - Figure Supplement 3B was conducted to test self-pupylation levels in the different Halo-PafA derivatives. For this purpose, we could use any Pup substrate such as SBP-sPup and SBPK4R-sPupK61R, instead of Ts-sPup and TS-sPupK61R, as they do not show any differences in pupylation activity. We chose Ts-sPup and TS-sPupK61R simply because any Pup substrates could be used for this purpose. Similarly, we did not need to incubate the reaction for a longer time to detect polypupylation, as our intention was to test “self-pupylation”. We demonstrated in Figure 1 – figure supplement 2 that polypupylation is dependent on the number or position of lysine residues in Pup substrate or tags. The results clearly showed that self-pupylation was almost completely abolished by the Halo8KR mutation. To clarify this, we added the following description in lines 168-169: “Ts-sPup and TS-sPupK61R were chosen as sPup substrates for this experiment, although any Pup substrates could have been used. The levels of self-pupylation were assessed.”

(3) Line 156: The statement that "the TS-tag completely abolished polypupylation in TS-sPup" is inaccurate. Using TSK8R-sPupK61R as the substrate, several bands appear, which likely represent Halo-PafA with varying degrees of polypupylation. Some bands also appear to correspond to those seen when using TS-sPup as a substrate. The authors should clarify how they distinguish between multipupylation and polypupylation in this case.

We sincerely appreciate your insight into clarifying the distinction between multipupylation and polypupylation. Polypupylation refers to the addition of a new Pup onto a previously linked Pup on the target protein, akin to polyubiquitination. In contrast, multipupylation involves multiple single pupylations at different positions on the target proteins. Since pupylation occurs exclusively at lysine residues in tag-Pup substrates, mutating all lysine residues to arginine, as in TSK48R-sPupK61R, prevents the mutant tag-Pup from linking to another Pup. This means that only single pupylation can proceed with this type of mutant Pup substrate. If multiple pupylated bands are observed with this mutant substrate, it indicates “multipupylation” rather than “polypupylation”, as shown in Figure 1-figure supplement 2D. The same applies to the pupylation bands in Figure 1-figure supplement 2E and F, as sSBP-sPupK61R and SBPK4R-sPupK61R lack lysine residues. By comparing these multipupylation bands, it is also possible to distinguish them from polypupylation bands, which are marked by yellow arrows. However, after 2-3 pupylation bands, higher-order bands become increasingly difficult to distinguish.

To clarify the mutation in the TS-tag, we revised the sentence in line 156 from “However, further mutations within the TS-tag completely abolished polypupylation in TS-sPup” to “However, further mutations of two lysine residues within the TS-tag, creating TSK8R-sPupK61R, completely abolished polypupylation in TS-sPup”. Additionally, we have inserted sentences in line 152 to define polypupylation and multipupylation, as described here.

(4) Line 160: Similar to the above concern about line 156, the claim that SBPK4R and sSBP completely prevented polypupylation is unconvincing and requires more supporting evidence.

Thank you for raising this concern. As mentioned above, both SBPK4R and sSBP lack lysine residues required for pupylation. As a result, these mutants can only undergo multiple single pupylations on the lysine residues of the target protein, which leads to “multipupylation”. In Figure 1-figure supplement 2E and F, pupylation bands by sSBP-sPupK61R or SBPK4R-sPupK61R do not display doublet bands (one from multipupylation and the other from polypupylation), as seen with SBP-sPup, marked by yellow arrows. Notably, Halo-PafA containing polypupylated branches migrates more slowly than one with an equal number of multipupylation events. To clarify this point, we have added the phrase “as shown in sSBP-sPupK61R and SBP4KR-sPupK61R” at the end of the sentence in line 160.

(5) Lines 176-177: The authors claim that PafAS126A exhibited reduced polypupylation compared to PafA, but given that PafAS126A may reduce depupylase activity, how could it reduce polypupylation levels? Moreover, it is hard to find any data supporting this conclusion in Figure 1 - Figure Supplement 3B.

We appreciate your insightful comment. At this point, we do not fully understand how the mutation that reduces depupylase activity also decreases polypupylation. It is possible that PafAS126A has a lower preference for pupylated Pup as a prey, which is required for polypupylation, since depupylase activity depends on recognizing pupylated Pup as a prey to remove it. Nonetheless, Halo-PafAS126A shows reduced levels of higher molecular weight bands compared to Halo-PafA, as shown in Figure 1-figure supplement 3B, while exhibiting increased pupylation in lower molecular weight bands, which represent either multipupylation or low-degree polypupylation. Since higher molecular weight bands (> 150 kD) are likely due to polypupylation, this result suggests reduced polypupylation and increased multipupylation in Halo-PafAS126A. To clarify this in the main text, we have added the following description in line 177: “as evidenced by the decreased levels of high molecular weight bands and an increase in low molecular weight bands”

(6) POST-IT system in cellulo validation: The system was developed using the Halo-tag, yet the in-cell validation uses FRB and FKBP instead, without explaining this switch. This inconsistency makes the logic of the experiment unclear.

We appreciate your insightful comment. The interaction between rapamycin and FRB or FKBP is known to be highly specific and robust, making this system useful in various biological contexts. Due to this property, rapamycin can induce interaction between two proteins when one is fused with FRB and the other with FKBP. Before testing or optimizing the POST-IT system in cells, we hypothesized that using the rapamycin-induced interaction between FRB and FKBP could introduce pupylation of the target protein, provided that PafA is fused with FRB or FKBP and the target protein is fused with the other. The results demonstrate that PafA can introduce pupylation of the target protein in a proximity-dependent manner via this chemically induced interaction. To further clarify this in the main text, we modified the original sentence in lines 214-216 as follows: “To mimic drug-target interaction-induced pupylation in live cells and assess the potential of PafA as a proximity-tagging system for target-ID, we incorporated the rapamycin-induced interaction between FRB and FKBP into our PL system, as this interaction between a small molecule and a protein is known to be highly specific and robust (Figure 3—figure supplement 1).”

(7) Line 209: The authors decided to use the SBP-tag for further studies due to better performance, but in Figure 3 - Figure supplement 1, they still used the unintroduced HB-Pup as the substrate, which is confusing and lacks explanation.

Thank you for raising your question. The SBP-tag is not superior to the TS-tag in terms of pupylation activity. However, the TSK8R mutant cannot bind to Strep-Tactin beads, while the SBP mutants, SBPK4R and sSBP, can bind to streptavidin. Therefore, we chose the SBP-tag instead of the TS-tag for further studies as a Pup substrate in POST-IT system, as we needed to pull down the target proteins. HB-Pup is consistently used as a control throughout various experiments, as it is the original Pup substrate. In Figure 3-figure supplement 1B and C, HB-Pup was used to test chemically induced pupylation by PafA. In these cases, it was not so critical which Pup substrate was chosen. Furthermore, we compared HB-Pup and different SBP-sPup substrates in Figure 3-figure supplement 1D, where HB-Pup was used as a control or for comparison. Although pupylation bands with HB-Pup appear more robust, this substrate contains multiple lysine residues, leading to high levels of polypupylation. To make it clear, we modified the sentence in line 209 to “Therefore, we decided to use the SBP-tag as a Pup substrate in the POST-IT system for further studies.”.

(8) Line 220: Both SBP-sPup and SBPK4R-sPupK61R are described as exhibiting efficient pupylation, but the data show mostly self-pupylation and little to no pupylation of the target protein.

Thank you for your concern. However, pupylation of the target protein is actually quite substantial, as the intensities of the free form and pupylated proteins are relatively similar, as shown in the upper panel of Figure 3-figure supplement 1D. Self-pupylation is always much higher than target pupylation, because PafA constantly pupylates itself, whereas pupylation of the target protein occurs only through interaction. Furthermore, V5-FRB-mKate2-PafA contains many lysine residues, which increases the levels of self-pupylation.

(9) Lines 222-224: The authors chose SBPK4R-sPupK61R to avoid polypupylation, although SBP-sPup did not cause detectable polypupylation. Neither substrate caused pupylation of the target protein, so the rationale behind this choice is unclear.

Thank you for raising your question. Similar to the above comment (#8), please refer to the pupylation bands of the target protein, as shown in the upper panel of Figure 3-figure supplement 1D. The pupylation band of the target protein is quite remarkable, as the intensities of the free form and pupylated proteins are comparable. Additionally, there are no multiple pupylation bands in either case, except for one additional weak multipupylation band, indicating no polypupylation by SBP-sPup, which does not have K-to-R mutations. Of course, SBPK4R-sPupK61R can only undergo single pupylation, as it does not contain lysine residues. Although we did not observe polypupylation by SBP-sPup in this experimental condition, it is possible that SBP-sPup may cause polypupylation under different experimental conditions or with other target proteins. Since SBPK4R-sPupK61R exhibits comparable pupylation of the target protein at least in this experiment setting as SBP-sPup, we selected SBPK4R-sPupK61R as the Pup substrate for POST-IT system to avoid any potential polypupylation that could be caused by SBP-sPup in other cases. We believe that polypupylation can introduce bias into the analysis and hinder the comprehensive discovery of additional target proteins for small molecules.

(10) Line 224: The authors conclude that rapamycin greatly reduced self-pupylation, but the supporting data are unclear.

Thank you for your constructive comments on our manuscript. Please refer to the lower panel of Figure 3-figure supplement 1D. When using either SBPK4R-sPupK61R or SBP-sPup, rapamycin treatment results in reduced levels of self-pupylation compared to the no-treatment control. However, we did not observe this reduction with HB-Pup and do not know the reason. To clarify this in the main text, we added the following description to the end of the sentence: “when using either SBPK4R-sPupK61R or SBP-sPup, as shown in the lower panel of Figure 3—figure supplement 1D”

(11) Line 234: The authors selected an 18-amino acid linker, but given that linkers longer than 10 amino acids enhance labeling, this choice should be explained.

Thank you for raising your question. In fact, a linker of 10 amino acids (aa) or longer is likely to behave similarly. We chose an 18 aa linker instead of a 40 aa linker primarily for the convenience of cloning and to reduce the potential for DNA sequence recombination associated with longer repeats. Additionally, a longer, flexible linker may behave like an intrinsically disordered protein (Harmon et al., 2017), which can lead to unwanted protein-protein interactions or phase separation. To elaborate on this, we added the following sentences after the sentence in line 233-235: “We chose the 18-amino acid linker instead of the 40-amino acid linker for easier cloning and to lower the risk of DNA recombination from longer repeats. Additionally, a longer, flexible linker may behave like an intrinsically disordered protein (Harmon et al., 2017), an unwanted feature for target-ID.”

(12) S126A and K172R mutations: The authors claim that these mutations additively enhanced pupylation under cellular conditions, but in Figure 3B, the band intensities appear similar for the wild-type and mutant versions.

Thank you for raising your concern. Although a single pupylation band appears similar among the three different Halo-PafA proteins, multipupylation bands are slightly but noticeably increased by the S126A and K172R mutations compared to Halo8KR-PafA. Since we used SBPK4R-sPupK61R as a Pup substrate, all higher molecular weight bands result from multipupylation rather than polypupylation. This illustrates why it is preferable to use SBPK4R-sPupK61R over SBP-sPup, as the pupylation bands with SBP-sPup are mixtures of poly- and multipupylation, making it difficult to assess levels of target labeling. To clarify this in the main text, we added the following description after the sentence in line 236: “as the higher molecular weight multipupylation bands are slightly but noticeably increased with these mutations compared to Halo8KR-PafA”

(13) Line 263: The authors selected DH5 for further experiments due to its efficiency, but the data suggest that the performance of DH1 to DH5 is similar.

We appreciate your question about the different dasatinib HTL derivatives. However, our data clearly show that DH2-5 derivatives bind significantly more effectively to Halo-PafA in vitro and in live cells compared to DH1 (Figure 4A and B). Additionally, the DH2-5 derivatives result in dramatically increased pupylation of the target protein in vitro and noticeable enhancement in live cells (Figure 4C and D). Among DH2 to DH5, there is no obvious difference in binding to Halo-PafA or pupylation of the target protein. Therefore, we chose DH5, as we believe that the longer linker in DH5 may facilitate the binding of a more diverse range of target proteins to dasatinib, enabling the discovery of additional target proteins.

(14) Line 309: The authors introduce HCQ and CQ as important drugs but then investigate the mechanism using DC661 without introducing or justifying the choice of this compound.

Thank you for your point. We explained the reason to choose DC661, a dimer form of CQ, instead of CQ for the synthesis of an HTL derivative in line 310. “assuming that a dimer would enhance binding affinity as previously described.” As the dimer forms of a drug or a small molecule such as testosterone dimers, estrogen dimers, and numerous anticancer drug dimers have been often developed to enhance drug effects (Paquin A et., Molecules 2021). Similarly, dimer forms of HCQ/CQ have been introduced and shown to be more potent (Hrycyna CA et al., ACS Chem Biol 2014; Rebecca VW et al., Cancer Discovery 2019). We expected that using a dimer form might offer higher probability to identify target proteins for HCQ/CQ.

(15) The authors suggest that multipupylation levels were enhanced but do not explain whether this might benefit the system or introduce other issues. Clarifying this point would provide valuable insight for potential users of this system.

Thank you for your thoughtful suggestion. Polypupylation likely leads to biased enrichment of a limited set of target proteins, and its levels may not correlate with the binding affinity of target proteins to the small molecule of interest, features that can negatively impact target-ID. In contrast, multipupylation may be correlated with binding affinity or interaction frequency, as we observed increased levels of multipupylation with higher Pup concentrations and longer incubation times. This suggests that target proteins with multiple lysines in proximity to PafA can be sequentially pupylated, starting with the most accessible lysine. However, if a target protein has only one accessible lysine, pupylation will occur only once, regardless of the protein’s affinity to the small molecule. In summary, while polypupylation may be a drawback for target-ID, multipupylation could be useful for both target-ID and understanding binding mode. To elaborate on this, we added the following additional explanation after the sentence in line 152: “, whereas multipupylation is more likely correlated with binding affinity or interaction frequency.”

(16) The author should address whether the Halotag ligand modification of the drug alters the binding properties between the drug and targets. That may be causing artifact binding of the drug and other proteins.

Thank you for your insightful comment. Yes, it is true that chemical modifications of the small molecule of interest, such as linker derivatization (e.g., HTL) or photo-affinity labeling, generally lead to reduced activity or affinity compared to the original molecule. Synthesizing a derivative is a common challenge across all target-ID methods, except for modification-free approaches, as we mentioned in the Discussion. However, modification-free methods like DARTS, CETSA, and TPP have their own limitations, including low sensitivity or high false positive rates. Identifying the optimal position for chemical modification on the small molecule of interest is critical. We chose dasatinib and HCQ/CQ as model compounds, because previous studies provided insights into their derivative synthesis. In addition, our data show that DH5 retains robust kinase inhibitory activity (Figure 4-figure supplement 2), and DC661-H1 exhibits potent autophagy inhibition (Figure 6-figure supplement 1). For novel compounds, a thorough structure-activity relationship study is essential to identify the optimal position for HTL derivative synthesis.

(17) The author stated there is no observable toxicity in zebrafish without providing a detailed analysis or enough data. Further analysis of the expression of Halo-PafA and its substrate sPup influence on toxicity or side effects to the living cells or animals would be needed. It is important for in vivo applications.

Thank you for your constructive suggestion. We have now included additional experimental data in Figure 7-figure supplement 1, showing no toxicity in zebrafish embryos expressing the POST-IT system. We assessed toxicity in two ways: by injecting the POST-IT DNA plasmid into one-cell-stage embryos for acute expression, and by using embryos from transgenic zebrafish expressing POST-IT under a heat-shock inducible promoter. Neither the injection nor the heat-shock activation of POST-IT expression resulted in any noticeable toxicity.